# Radiance-based Retrieval Bias Mitigation for the MOPITT Instrument: The Version 8 Product

Merritt N. Deeter[1], David P. Edwards[1], Gene L. Francis[1], John C. Gille[1], Debbie Mao[1], Sara Martínez-Alonso[1], Helen M. Worden[1], Dan Ziskin[1], and Meinrat O. Andreae[2,3]

[1]Atmospheric Chemistry Observations and Modeling Laboratory, National Center for Atmospheric Research, Boulder, CO, USA
[2]Max Planck Institute for Chemistry, P.O. Box 3060, D-55020 Mainz, Germany
[3]Scripps Institution of Oceanography, University of California San Diego, La Jolla, CA 92093-0221, USA

**Correspondence:** M. N. Deeter (mnd@ucar.edu)

**Abstract.**

The MOPITT ("Measurements of Pollution in the Troposphere") satellite instrument has been making nearly continuous observations of atmospheric carbon monoxide (CO) since 2000. Satellite observations of CO are routinely used to analyze emissions from fossil fuels and biomass burning, and the atmospheric transport of those emissions. Recent enhancements to the MOPITT retrieval algorithm have resulted in the release of the Version 8 (V8) product. V8 products benefit from updated spectroscopic data for water vapor and nitrogen used to develop the operational radiative transfer model and exploit a new method for minimizing retrieval biases through parameterized radiance bias correction. In-situ datasets used for algorithm development and validation include the NOAA ("National Oceanic and Atmospheric Administration") and HIPPO ("HIA-PER Pole-to-Pole Observations") datasets used for earlier MOPITT validation work in addition to measurements from the ACRIDICON-CHUVA ("Aerosol, Cloud, Precipitation, and Radiation Interactions and Dynamics of Convective Cloud Systems - Cloud processes of the main precipitation systems in Brazil: A contribution to cloud resolving modeling and to the GPM (Global Precipitation Measurement)"), KORUS-AQ ("The Korea-United States Air Quality Study") and ATom ("The Atmospheric Tomography Mission") programs. Validation results illustrate clear improvements with respect to long-term bias drift and geographically variable retrieval bias. For example, whereas bias drift for the V7 TIR-only product exceeded 0.5 %/yr for levels in the upper troposphere (e.g., at 300 hPa), bias drift for the V8 TIR-only product is found to be less than 0.1 %/yr at all levels. Also, whereas upper-tropospheric (300 hPa) retrieval bias in the V7 TIR-only product exceeded 10 % in the tropics, corresponding V8 biases are less than 5 % (in terms of absolute value) at all latitudes and do not exhibit a clear latitudinal dependence.

## 1 Introduction

MOPITT ("Measurements of Pollution in the Troposphere") is a gas correlation radiometer instrument on the NASA Terra satellite which permits retrievals of CO vertical profiles using both thermal-infrared (TIR) and near-infrared (NIR) measurements. The MOPITT instrument has been in operation since 2000 (Drummond et al., 2016) resulting in a long-term data record

well suited for a variety of applications. Satellite measurements of CO are used in air quality forecasts (Inness et al., 2015) as well as to estimate CO emissions (e.g., Kopacz et al. (2010); Jiang et al. (2017); Zheng et al. (2018)) and to study how fires and urban pollution influence atmospheric chemistry on regional to global scales (e.g., Heald et al. (2003); Edwards et al. (2004); Pfister et al. (2005); Turquety et al. (2007); Shindell et al. (2006); Emmons et al. (2010); Kumar et al. (2013); Gaubert

et al. (2017); Miyazaki et al. (2018)). Since its launch, MOPITT retrieval products have improved continuously as the result of accumulated knowledge regarding the instrument, radiative transfer modeling methods, and geophysical variables (Worden et al., 2014; Deeter et al., 2017). Moreover, the availability of new in-situ datasets (e.g., from field campaigns) has enabled the quantitative analysis of MOPITT retrieval biases in a variety of contexts. As illustrated below, such in-situ datasets may be used to both (1) mitigate temporally and geographically variable retrieval biases and (2) independently validate the resulting

CO product.

MOPITT retrieval products are generated with an iterative optimal estimation-based retrieval algorithm which involves both the MOPITT calibrated radiances and a priori knowledge of CO variability (Deeter et al., 2003). CO retrievals of log(VMR) are performed on a retrieval grid with ten pressure levels (surface, 900 hPa, 800 hPa, ..., 100 hPa). Retrieval layers, used internally in the MOPITT retrieval algorithm, are defined by the layers between each level in this grid and the next-highest level in the

grid (Francis et al., 2017). For example, the surface-level retrieval product actually represents the mean volume mixing ratio for the layer between the surface and 900 hPa. Retrieved CO total column values are calculated directly from the CO profile and are thus not retrieved independently. A priori CO profiles are derived from a model climatology, and vary seasonally and geographically; the a priori climatology introduced for processing MOPITT Version 7 products (Deeter et al., 2017) is unchanged for the new Version 8 products described below.

All MOPITT CO retrievals are based on a defined subset of the Average (A) and Difference (D) radiances from MOPITT channels 5, 6, and 7; each channel is associated with a specific TIR or NIR gas correlation radiometer (Drummond et al., 2010). TIR-only retrievals are based on the 5A, 5D, and 7D radiances in the 4.7 $\mu$m band, whereas NIR-only retrievals are based solely on the ratio of the 6D and 6A radiances in the 2.3 $\mu$m band. MOPITT TIR-only retrievals are typically most sensitive to CO in the mid- and upper-troposphere, except in scenes characterized by strong thermal contrast (Deeter et al., 2007). MOPITT

NIR-only retrievals are most useful for retrievals of CO total column (Deeter et al., 2009; Worden et al., 2010). Unique "multi-spectral" TIR-NIR retrievals exploit the 5A, 5D, 7D, 6D, and 6A radiances. The TIR-NIR product offers greater vertical resolution than TIR-only or NIR-only products, and features the greatest sensitivity to CO in the lower troposphere (Deeter et al., 2013). However, because NIR measurements rely on reflected solar radiation, the main benefits of the TIR-NIR product are limited to daytime MOPITT observations over land.

Overall retrieval biases for MOPITT V7 products were previously shown to be less than about 5% at all retrieval levels for the TIR-only, NIR-only, and TIR-NIR retrievals (Deeter et al., 2017). However, over the MOPITT mission, validation results for V7 also indicated significant long-term trends in the retrieval biases, i.e., bias drift. For example, V7 TIR-only biases at the 800 and 400 hPa retrieval levels exhibited bias drift of -0.41 %/yr and 0.81 %/yr, respectively. However, opposing drift in the upper and lower troposphere appears to mostly cancel with respect to the retrieved total column (Deeter et al., 2013).

Similar drift values for 800 hPa and 400 hPa retrievals are indicated in the updated bias timeseries plots for the V7 TIR-only

product (based on in-situ profiles acquired from 2000-2018) shown in Fig. 1; validation methods are described in Section 3. Considerations of long-term drift are particularly important for satellite-based analyses of CO long-term trends (Worden et al., 2013).

Moreover, as indicated in Fig. 2, validation results using the HIPPO ("HIAPER Pole-to-Pole Observations") dataset (Wofsy et al., 2011) revealed significant latitudinal variability in the V7 TIR-only retrieval biases (Deeter et al., 2017). In the upper troposphere, for example, V7 TIR-only retrieval biases are much larger in the Tropics than outside the Tropics. This effect might, for example, be related to the modeling of water vapor absorption in the MOPITT TIR passband (Edwards et al., 1999) or perhaps to the accuracy of water vapor fields which are used in the MOPITT retrieval algorithm (Pan et al., 1995; Wang et al., 1999). V7 TIR-only retrieval biases derived for the HIPPO dataset are plotted versus water vapor total column (derived from the MERRA-2 reanalysis) in Fig. 3. The figure indicates that retrieval biases increase with increasing water vapor in the upper troposphere but exhibit the opposite behavior in the lower troposphere. Correlation coefficients for this dependence are largest in the upper troposphere (0.67 at 200 hPa) and lower troposphere (-0.42 at 900 hPa).

The remaining sections of this manuscript describe revisions made to the MOPITT Version 8 retrieval algorithm (Section 2), the validation methodology and results (Section 3) and, finally, the conclusions drawn from the results (Section 4).

## 2    Version 8 Retrieval Algorithm Enhancements

As detailed in the sections below, the Version 8 retrieval algorithm incorporates updated spectroscopic information used in the radiative transfer model, improved methods for radiance bias correction and averaging kernel calculations and, finally, the most recent version of the MODIS cloud mask.

### 2.1    Radiative Transfer Modeling

The MOPITT operational retrieval algorithm relies on a fast radiative transfer model, known as MOPFAS, to simulate measured radiances for specified atmospheric and surface conditions (Edwards et al., 1999). Within the MOPITT TIR and NIR filter passbands, measured radiances are sensitive to atmospheric concentrations of several gases including CO, water vapor, and $N_2O$. Accurate spectroscopic data (e.g., absorption line strengths and line widths) are needed for each of these gases in the development of MOPFAS. For most gases, the V8 operational radiative transfer model is based on the HITRAN2012 spectral database (Rothman et al., 2013), which is the same version of HITRAN used for MOPITT V7 processing. However, spectroscopic data for water vapor and nitrogen have each been updated for V8. Water vapor continuum absorption in the V8 operational radiative transfer model is based on version 3.2 of the MT_CKD spectroscopic model from AER (Mlawer et al., 2012) whereas earlier MOPITT radiative transfer models employed version 1.0. The radiative effects of molecular nitrogen, which were not previously represented in MOPFAS, are now derived from the line-by-line model GENLN3 (Edwards, 1992). Absorption by nitrogen in the MOPITT TIR band occurs through collisions between $N_2$ molecules (Richard et al., 2012). A comparison of top-of-atmosphere TIR transmittances for the two models for water vapor continuum absorption, along with the collisionally-induced transmittance of nitrogen, is shown in Fig. 4. For water vapor, transmittance differences exceeding 10%

are observed for the two models. The figure also shows that the collisionally-induced transmittance of nitrogen varies between approximately 1 and 10% across the MOPITT TIR passband. With respect to radiances integrated over the MOPITT TIR passband, model calculations indicate that the inclusion of both the new water vapor model and collisionally-induced nitrogen absorption each produce changes of several percent. In addition, to reflect current tropospheric conditions, the globally-fixed concentration of $CO_2$ assumed in the V8 operational radiative transfer model was increased to 410 ppm. This change was found to produce a negligible effect (less than 0.05%) on the MOPFAS-simulated radiances, however.

## 2.2 Radiance Bias Correction

The MOPITT Level 2 processor exploits radiance bias correction factors to compensate for relative biases between simulated radiances calculated by MOPFAS and actual calibrated Level 1 radiances from the instrument. Without some form of compensation, radiance biases would produce large biases in the retrieved CO profiles. Radiance bias correction factors compensate for a variety of potential bias sources including errors in instrumental specifications, forward model errors related to the development of MOPFAS, errors in assumed spectroscopic data, and geophysical errors (Deeter et al., 2014). For previous MOPITT products, radiance bias correction factors were determined empirically by minimizing retrieval biases estimated from validation using aircraft in-situ profiles (Deeter et al., 2017). This method resulted in radiance bias correction factors for each of the TIR and NIR radiances used in the MOPITT CO retrieval products. Within the retrieval algorithm, these correction factors are applied by scaling the simulated radiances produced by MOPFAS each time it is executed.

For V8 processing, radiance bias correction is based on a new parameterization involving both (1) the date of the MOPITT observation and (2) the water vapor total column at the time and geographic location of the MOPITT observation, as derived from the MERRA-2 water vapor profiles needed to execute MOPFAS (Deeter et al., 2017). This strategy is based on the empirical linear dependence of MOPITT retrieval biases on time and water vapor, as shown in Section 1. Although the use of this parameterization does not depend on an exact understanding of the physical origin of the biases, it does assume that radiance biases vary linearly with respect to both time and vertically integrated atmospheric water vapor. Within the retrieval software, "dynamic" radiance bias correction factors for V8 are therefore calculated using the relation

$$R^i = R_0^i + R_t^i N_{dys} + R_w^i WV \tag{1}$$

where $R^i$ is the radiance correction factor to be applied to the model-simulated value for radiance $i$, $N_{dys}$ is the number of elapsed days since January 1, 2000, $WV$ is the water vapor total column (or "precipitable water vapor", expressed in molecules/cm$^2$) determined from the MERRA-2 reanalysis (temporally and spatially interpolated to the time and location of the MOPITT observation), and $R_0$, $R_t$, and $R_w$ are the empirically-determined parameters which effectively minimize overall retrieval bias, bias drift and bias water vapor sensitivity. Values of $R_0$, $R_t$, and $R_w$ for the 5A, 5D, 6D, and 7D radiances used for V8 operational processing are listed in Table 1. Values of these parameters for the 5A, 5D and 7D radiances (i.e., the TIR radiances) were obtained by minimizing retrieval biases for the TIR-only validation results. Values of these parameters for the 6D radiance were obtained by minimizing retrieval biases using NIR-only validation results. Experiments performed to optimize $R_t$ were strictly based on the analysis of bias drift determined using the NOAA profile set; experiments performed

to optimize $R_w$ were based solely on the analysis of water vapor-dependent biases determined using the HIPPO profile set (described further in Section 3). Optimized values of $R_0$ were based on both the NOAA and HIPPO results. Non-zero values for the time-dependent term, $R_t$, were found to be necessary only for the 6D and 7D radiances. Non-zero values for $R_w$ were found to be necessary only for the 5D and 7D radiances.

## 2.3 Averaging Kernel Calculations

The averaging kernel matrix $A$ quantifies the sensitivity of the retrieved profile to the true profile and is provided as a diagnostic for each retrieval in all MOPITT products. For users interested in comparisons of MOPITT retrieved CO total column values with other datasets (or model output), the V8 Level 2 product files also include the total column averaging kernel $a$. The vector $a$ quantifies the sensitivity of the retrieved total column to perturbations at each level of the CO profile, as described in the MOPITT User's Guide (MOPITT Algorithm Development Team, 2018). If $C_{rtv}$ is the retrieved CO total column, and $x$ is the state vector comprised of actual CO log(VMR) values, the total column averaging kernel element for profile level $j$ is defined by

$$a_j = \partial C_{rtv}/\partial x_j \tag{2}$$

Given a comparison CO profile $x_{cmp}$ (typically either from in-situ measurements or model output), the total column averaging kernel vector may be used to simulate MOPITT total column retrievals using the equation

$$C_{sim} = C_a + a(x_{cmp} - x_a) \tag{3}$$

where $C_a$ is the a priori total column value corresponding to the a priori profile $x_a$. (Both $C_a$ and $x_a$ are provided for each retrieval in the V8 Level 2 product files.)

For V8, the method of calculating $a$ has been revised for consistency with the method of Rodgers (2000), Section 4.3. Specifically, the total column averaging kernel is now calculated as

$$a^T = h^T A \tag{4}$$

where $h$ is the partial column operator (i.e., the vector of derivatives of CO partial column values with respect to perturbations in log(VMR), referenced to the retrieved profile) and $A$ is the full averaging kernel matrix. The vector $h$ is calculated internally in the MOPITT retrieval code. While the new method for calculating $a$ is more rigorous than the previously used method, resulting differences in total column validation statistics (correlation coefficient, bias, and standard deviation) were found to be insignificant. In addition, a software coding error has been corrected which, for V7 products, resulted in erroneous values of $a$ for retrieved profiles with less than 10 valid levels (i.e., surface pressures less than 900 hPa).

## 2.4 Cloud Detection

The MOPITT retrieval algorithm generally discards MOPITT observations in which clouds are detected. The cloud detection algorithm used for this purpose makes use of both the MOPITT radiances and the MODIS cloud mask (Francis et al., 2017).

The MOPITT Channel 7 Average radiance is employed for cloud detection since it is relatively less sensitive to CO variability than the other MOPITT TIR radiances. For V8, two changes have been made with respect to cloud detection. First, V8 products for the entire MOPITT mission are produced using MODIS Collection 6.1 cloud mask files. (Differences between Collection 6 and Collection 6.1 MODIS cloud products are documented in Moeller and Frey (2017)). Second, due to changes in the radiative transfer model described in Sec. 2.1, the threshold ratio value used to test the MOPITT Channel 7 Average radiance for cloudiness (relative to the MOPFAS-based first-guess value) was increased from 0.955 to 1.000. This value was selected to achieve consistent rates of clear-sky determinations for V7 and V8 processing; hence, any observed differences between V7 and V8 products are unlikely to be related to cloud detection.

## 3 Validation

Below, retrieval validation results are reported separately for V8 TIR-only, NIR-only and TIR-NIR products. Validation results are based on statistical comparisons of MOPITT retrieval products (CO volume mixing ratio profiles and total columns) with in-situ vertical profiles measured from aircraft. For this purpose, in-situ measurements are assumed to be exact and representative of an extended region around the sampling location. Other remote sensing datasets, such as the TCCON ('Total Carbon Column Observation Network') and NDACC ('Network for the Detection of Atmospheric Composition Change') datasets are potentially useful for MOPITT validation (Buchholz et al., 2017), however results are more difficult to analyze due to differences in retrieval averaging kernels and a priori (Rodgers, 2003). Thus, results presented in this paper are solely based on aircraft in-situ profiles whereby averaging kernel effects are taken fully into account.

Because of the coarse vertical resolution of the radiance weighting functions (or "Jacobians") and the underconstrained nature of the retrieval process, retrieval products obtained with optimal estimation-based algorithms are constrained by a priori information as well as the measurements (Pan et al., 1998; Rodgers, 2000). A priori information is represented by (1) an a priori profile $x_a$ and (2) an a priori covariance matrix, which defines the strength of the a priori constraint. The relationship between the true profile $x_{true}$, $x_a$, and retrieved profile $x_{rtv}$ is expressed by the equation

$$x_{rtv} = x_a + A(x_{true} - x_a) \tag{5}$$

where $A$ is the retrieval averaging kernel matrix. The vector quantities $x_{true}$, $x_a$ and $x_{rtv}$ are expressed in terms of log(VMR) rather than VMR itself.

## 3.1 Validation Datasets

V8 validation results reported below exploit a large set of CO vertical profiles measured by the NOAA Global Monitoring Division using an airborne flask-sampling system followed by laboratory analysis (Sweeney et al., 2015). Typical profiles are derived from a set of twelve flasks. Reproducibility of the laboratory-measured CO dry-air mole fractions, which are measured by either a vacuum UV–resonance fluorescence spectrometer or a reduction gas analyzer is better than 1 ppb. This set is composed of profiles obtained during flights at 21 fixed sites (mainly over North America) between 2000 and 2018.

The consistency and high accuracy characterizing this set of profiles is the primary basis for quantifying long-term changes in MOPITT retrieval biases (Deeter et al., 2003, 2017).

CO vertical profiles acquired from aircraft-based instruments during field campaigns complement the NOAA set of profiles. Pertinent characteristics of these datasets (and the NOAA dataset) are listed in Table 2. The HIPPO field campaign was conducted in five phases between 2009 and 2011 (Wofsy et al., 2011) and has been especially useful for MOPITT validation (Deeter et al., 2014, 2017). Flights for HIPPO were conducted during January 2009 (Phase 1), November 2009 (Phase 2), April 2010 (Phase 3), June 2011 (Phase 4), and August-September 2011 (Phase 5). Because of the wide range of latitudes represented, this set of CO vertical profiles is useful for investigating geographically variable retrieval biases.

Validation results are also reported below for several more recent field campaigns, including (1) ACRIDICON-CHUVA ("Aerosol, Cloud, Precipitation, and Radiation Interactions and Dynamics of Convective Cloud Systems - Cloud processes of the main precipitation systems in Brazil: A contribution to cloud resolving modeling and to the GPM (Global Precipitation Measurement)") (Wendisch et al., 2016), hereafter referred to as "AC", (2) KORUS-AQ ("The Korea-United States Air Quality Study", https://espo.nasa.gov/korus-aq/content/KORUS-AQ_White_Paper), and (3) ATom ("The Atmospheric Tomography Mission", https://espo.nasa.gov/atom/content/ATom). CO measurements from AC were previously used to validate the MOPITT V6 product (Deeter et al., 2016), however CO datasets from KORUS-AQ and ATom are used here for MOPITT validation for the first time. AC was conducted in September 2014, while the KORUS-AQ campaign was conducted from April to June, 2016. For the ongoing ATom campaign, results are presented for Phases 1 and 2, conducted in July and August, 2016, and January and February, 2017, respectively. CO measurements for AC were performed using an Aero-Laser 5002 vacuum UV resonance fluorescence instrument (Wendisch et al., 2016), CO measurements for KORUS-AQ were performed with the DACOM ("Differential Absorption Carbon monOxide Measurement") instrument (Sachse et al., 1987) and CO measurements for both HIPPO and ATom were performed with the QCLS ("Quantum Cascade Laser Spectrometer") instrument (Santoni et al., 2014).

For matching MOPITT retrieved profiles with in-situ profiles, a maximum collocation radius of 50 km was employed for the NOAA and KORUS-AQ profiles whereas a value of 200 km was used for the HIPPO, AC and ATom profiles. The smaller radius for NOAA and KORUS-AQ was chosen to reduce validation errors resulting from large horizontal CO gradients associated with urban emissions from North America and Korea. For all in-situ datasets, a maximum of 12 hours was allowed between the time of the MOPITT observation and sampling time of the in-situ data. Maximum altitudes for individual profiles varied in the datasets from approximately 7 to 14 km. In order to obtain a complete validation profile for comparison with MOPITT retrievals, each in situ profile was extended vertically above the highest-altitude in-situ measurement using the CAM-chem (Community Atmosphere Model with Chemistry) chemical transport model (Lamarque et al., 2012) and then resampled to the standard pressure grid used for the MOPITT operational radiative transfer model (Martínez-Alonso et al., 2014). Validation results for the MOPITT 100 hPa retrieval level are not reported below, since apparent retrieval errors due to reliance on the model-based extension at the top of the profile is much greater than for lower retrieval levels. Reliable validation of the MOPITT 100 hPa retrieval level will require in-situ profiles that reach higher altitudes than are currently available.

V8 validation results are separately reported below for two groups of in-situ profiles. Validation results for the NOAA and HIPPO datasets, representing the first class of profiles, indicate the success of the retrieval bias minimization method described in Section 2, which was specifically optimized to minimize retrieval biases for these two datasets. In addition, validation results are presented for in-situ profiles acquired during the three campaigns (AC, KORUS-AQ and ATom) which were not exploited in the development of the bias minimization method. The results for this second class of profiles provide a completely independent evaluation of the MOPITT retrieval biases.

## 3.2 TIR-only

Validation results derived from the NOAA aircraft flask samples for the V8 TIR-only products are presented as a retrieval bias timeseries plot in Fig. 5. Corresponding results for the V7 TIR-only products were shown in Fig. 1. Each panel in the figure corresponds to a particular MOPITT retrieval level. (Results are not shown for the 100 hPa retrieval level, since the corresponding layer is generally not well measured in the aircraft in-situ datasets.) Retrieval biases for individual overpasses are quantified as relative deviations (expressed in percent) between the mean retrieved log(VMR) value and the corresponding value calculated using the in-situ profile data, a priori profile, and MOPITT averaging kernel matrix (Deeter et al., 2017). Validation statistics for total column and alternating retrievals levels, including relative bias, standard deviation, and correlation coefficient, are also summarized and compared with statistics for V7 products in Table 3. Correlations due simply to the variability of the a priori are avoided by basing correlation coefficient calculations on $(x_{rtv} - x_a)$ rather than $x_{rtv}$.

Overall biases for V8 TIR-only products for the NOAA profile set (averaged over all validation sites and over the whole mission) vary from -1.3 to 3.0 %. These small values are roughly similar to values for the V7 TIR-only product. Bias drift, however, is significantly improved for V8. Bias drift is calculated as the slope of a least-squares best fit applied to the timeseries data presented in Fig. 5, and converted from units of $\Delta$log(VMR)/yr to %/yr as described in Deeter et al. (2017). For each retrieval level depicted in Figure 5, the best-fit slope of the timeseries is within the uncertainty of the slope derived from the least-squares fit, which is generally less than 0.1%/yr at all retrieval levels. These results demonstrate that retrieval bias drift has been effectively eliminated in V8 TIR-only products at all retrieval levels, at least over North America. The lack of long-term CO aircraft-based datasets for other regions prevents general conclusions regarding the geographical variability of bias drift in V8 products.

V8 TIR-only biases derived from HIPPO validation results are plotted versus latitude in Fig. 6 and against water vapor total column in Fig. 7. Bin-averaged latitude-dependent biases for V8T are compared with V7T results in Table 4. Clear improvements are apparent in comparison with the V7 results presented in Figs. 2 and 3, in both the upper and lower troposphere. For example, at 300 hPa, bin-averaged latitude-dependent biases for V7T vary from about -4 to 12%, in comparison to -3 to 5% for V8T. This improvement is consistent with the weaker dependence of biases on water vapor, as indicated by comparing Figs. 3 and 7. At 300 hPa, the bias water vapor dependence (determined again by a least-squares fit) decreases from $1.27 \times 10^{-22}$ %/mol/cm$^2$ for V7T to $3.25 \times 10^{-23}$ %/mol/cm$^2$ for V8T. Significant improvements are also apparent in Figs. 6 and 7 in the lower troposphere, e.g., at 800 hPa.

V8 TIR-only biases as determined from validation results for the AC, KORUS-AQ, and ATom campaigns are plotted versus latitude in Fig. 8. Validation statistics for the individual campaigns are also listed in Table 5. With a few exceptions, the V8T latitudinally bin-averaged retrieval biases for both the HIPPO validation results in Fig. 6 and for the three other campaigns in Fig. 8 are generally within ± 5% at all retrieval levels. Biases outside of this range are most evident between 60 N and 90 N. However, this could be related to the sparseness of profiles in this region and the influence of a small number of outliers.

## 3.3 NIR-only

A timeseries plot for V8 NIR-only retrieval biases based on the NOAA profile set is presented in Fig. 9. Statistics are also summarized and compared with V7 validation statistics in Table 3. Whereas mean V7N biases for the NOAA profile set exceeded 2%, overall biases for V8N are less than 1%. With respect to long-term stability, NIR-only bias drift has been reduced from about -0.3%/yr to less than 0.1%/yr at all levels, which again is comparable to the bias drift statistical uncertainty. MOPITT NIR-only retrievals are only produced for daytime observations over land. Thus, since the HIPPO campaign was primarily conducted over the Pacific Ocean, profiles from that campaign are not useful for validating MOPITT NIR-only retrievals. V8N retrieval biases for the AC and KORUS-AQ campaigns are, however, plotted versus latitude in Fig. 10; corresponding statistics are also summarized in Table 5. V8N biases based on the KORUS-AQ profiles are close to 1%, which is consistent with the NOAA profile results, whereas biases for the AC campaign are close to 5%. For V8N products, the lack of aircraft profiles over land for regions other than North America, South Korea, and the Amazon Basin prevents a full analysis of retrieval bias geographical variability.

## 3.4 TIR-NIR

V8 TIR-NIR retrieval biases based on the NOAA profiles are presented as timeseries plots in Fig. 11, with summary statistics compared to statistics for V7 in Table 3. As indicated in that Table and in various figures, retrieval biases and standard deviations tend to be larger for both the V7 and V8 TIR-NIR products compared to the corresponding TIR-only products. This effect results from the strategy to amplify the weight assigned to the NIR measurements in the NIR-only and TIR-NIR products (Deeter et al., 2012). Overall biases for V8, which vary from about -5% at 600 hPa to about 7% at 200 hPa, are slightly larger than for V7. Bias drift, however, which exhibits large values for both the lower and upper troposphere for V7 products (e.g., 0.94%/yr at 800 hPa and 1.29%/yr at 200 hPa), has been decreased to statistically negligible values (0.1%/yr or less) at all retrieval levels for V8. V8 TIR-NIR retrieval biases for the AC, KORUS-AQ, and ATom campaigns are plotted versus latitude in Fig. 12. Results shown in Fig. 12 represent both land scenes (where both TIR and NIR radiances are exploited) and ocean scenes (where retrievals are based only on TIR radiances). Summary statistics for the AC and KORUS-AQ campaigns are also listed in Table 5. V8 TIR-NIR biases for the AC and KORUS-AQ campaigns are generally consistent with the corresponding NOAA results (within 5% at all levels).

## 4 Conclusions

Approaching a length of two decades, the MOPITT record for tropospheric CO is uniquely qualified for studies of both climate and air quality. However, the application of satellite-based datasets in climate studies strictly requires that those datasets be unaffected by bias variability which could be misinterpreted as a geophysical (climate-related) signal. Thus, the systematic comparison of satellite datasets to long-term in-situ datasets, as demonstrated above, is an essential prerequisite to the use of satellite data in climate studies.

Drifting retrieval bias evident in earlier MOPITT products could be the result of long-term instrumental degradation and/or temporal inconsistencies in the MERRA (and MERRA-2) temperature and water vapor profiles assumed in the MOPITT retrieval algorithm. Instrumental effects might include, for example, degradation in the sensors measuring the gas correlation cell temperatures and pressures and/or long-term composition changes within the gas correlation cells. The newest release of the MOPITT product, Version 8, incorporates an improved radiance bias correction method which sharply decreases both retrieval bias drift and bias geographical variability. The method involves a simple linear parameterization relating radiance biases in both the TIR and NIR channels to (1) the number of elapsed days since the beginning of the mission and (2) the vertically integrated water vapor at the location of the satellite measurement. The new MOPITT V8 product also benefits from updated spectroscopic data for water vapor and nitrogen.

Our chosen strategy to rely on the stability of the NOAA aircraft in-situ data to determine the time-dependent radiance bias correction factors does imply that the fidelity of CO trends in the radiance bias-corrected MOPITT data (i.e., the V8 retrieval product) is ultimately limited by the stability of the NOAA in-situ measurements. However, the NOAA measurements are widely accepted as a standard for long-term CO analyses, and are calibrated using the World Meteorological Organization (WMO) mole fraction scale (Sweeney et al., 2015). The NOAA dataset is therefore well qualified for climate analyses. On the other hand, we acknowledge that bias drift in the V8 product for regions not represented by the NOAA aircraft network (primarily covering North America) could be substantially different. This seems unlikely though, particularly if the source of the bias is instrumental.

Validation results demonstrate that MOPITT V8 products, including TIR-only, NIR-only and TIR-NIR variants, are typically characterized by biases of less than about 5%, with bias drift generally less than 0.1% $yr^{-1}$. Geographically variable biases have also been substantially reduced compared to V7 products. Validation results for the AC, KORUS-AQ and ATom campaigns, which were not involved in the development of the radiance bias correction method, are consistent with results for the NOAA and HIPPO profile sets. Ongoing validation work will include systematic comparisons of MOPITT products with other satellite products as well as ground-based remote sensing instruments.

*Data availability.* MOPITT Version 8 products are freely available through NASA's EarthData portal at https://earthdata.nasa.gov/. NOAA in situ CO profiles are available at http://www.esrl.noaa.gov/gmd/ccgg/aircraft/index.html. In situ CO data from the HIPPO field campaign are available at http://hippo.ornl.gov/dataaccess. CO in-situ data from the QCLS instrument during the ATom campaign are archived and publicly accessible from https://espoarchive.nasa.gov/archive/browse/atom/DC8/QCLS-CH4-CO-N2O. CO in-situ data from the DACOM instrument

during the KORUS-AQ campaign are archived and publicly accessible from https://www-air.larc.nasa.gov/cgi-bin/ArcView/korusaq. The full data set from the ACRIDICON-CHUVA campaign is archived and publicly accessible from the HALO database maintained by the German Aerospace Center (DLR) at https://halo-db.pa.op.dlr.de/mission/5.

*Author contributions.*  MD led the development and validation of the MOPITT Version 8 product and prepared the figures and manuscript. HW, DE and GF guided the changes made to the MOPITT radiative transfer model. SM managed the validation datasets. JG provided expertise with the MOPITT instrument. DM and DZ provided software engineering and data management support. MA provided measurements from the ACRIDICON-CHUVA experiment and assisted in their interpretation. All authors discussed the results and contributed to the final manuscript.

*Competing interests.*  The authors declare that they have no conflict of interest.

*Acknowledgements.*  The NCAR MOPITT project is supported by the National Aeronautics and Space Administration (NASA) Earth Observing System (EOS) Program. The National Center for Atmospheric Research (NCAR) is sponsored by the National Science Foundation.

We thank the ACRIDICON-CHUVA team for their cooperation during the AC campaign and the government of Brazil for permission to conduct the HALO flights. We acknowledge the support of the ACRIDICON-CHUVA campaign by the Max Planck Society, the German Aerospace Center (DLR), FAPESP (São Paulo Research Foundation), and the German Science Foundation (Deutsche Forschungsgemeinschaft, DFG) within the DFG Priority Program (SPP 1294) "Atmospheric and Earth System Research with the Research Aircraft HALO (High Altitude and Long Range Research Aircraft)".

Thanks are due also to the HIPPO and ATom Science Teams including Bruce Daube, Roisin Commane, Eric Kort, Greg Santoni, and Steve Wofsy and to the KORUS-AQ Science Team, particularly Glenn Diskin.

Finally, we also thank Colm Sweeney in NOAA's Global Monitoring Division for providing CO in-situ profiles from the Global Greenhouse Gas Reference Network Aircraft Program.

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

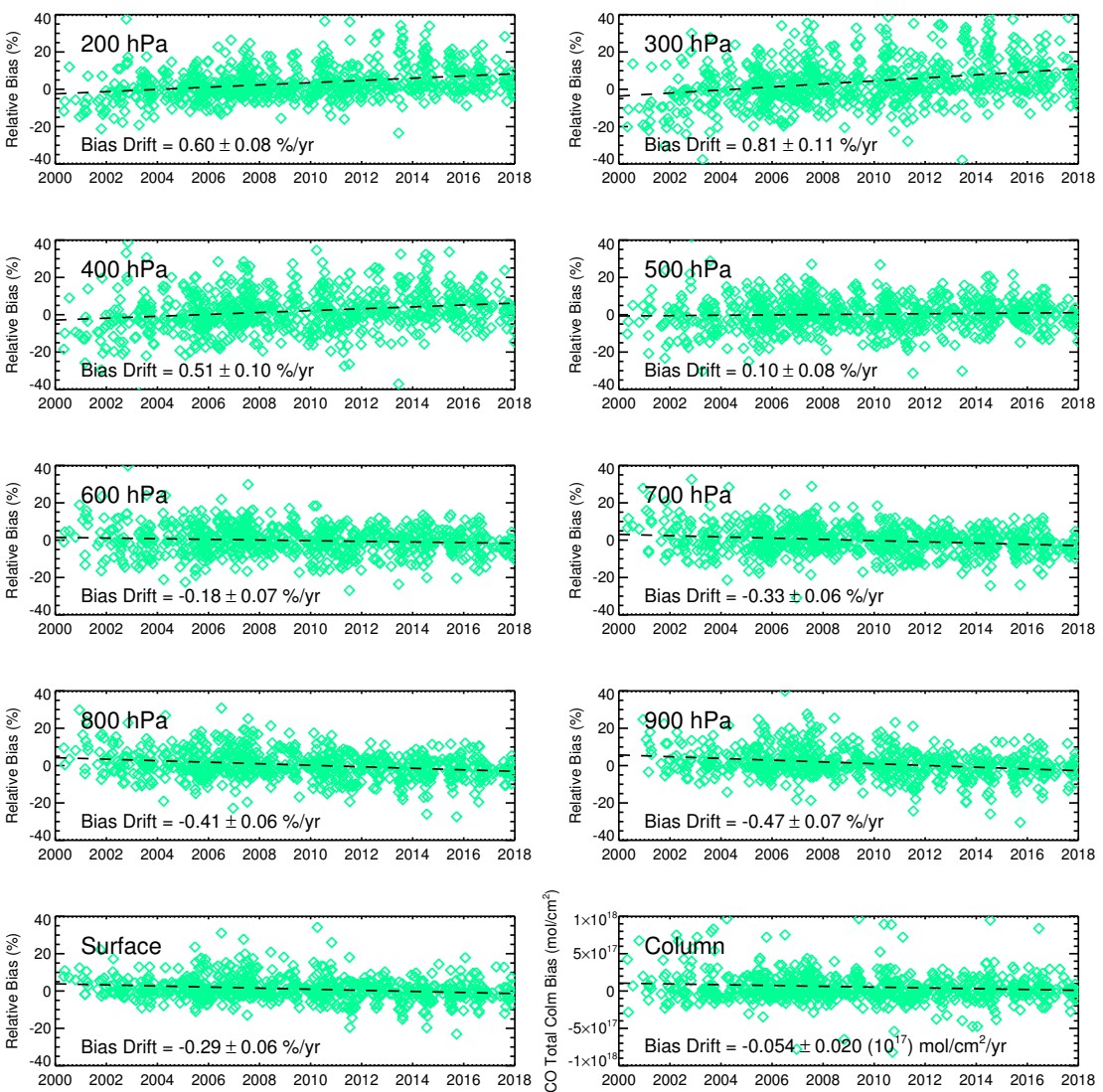

**Figure 1.** Retrieval bias drift for V7 TIR-only products based on the NOAA flask measurements.

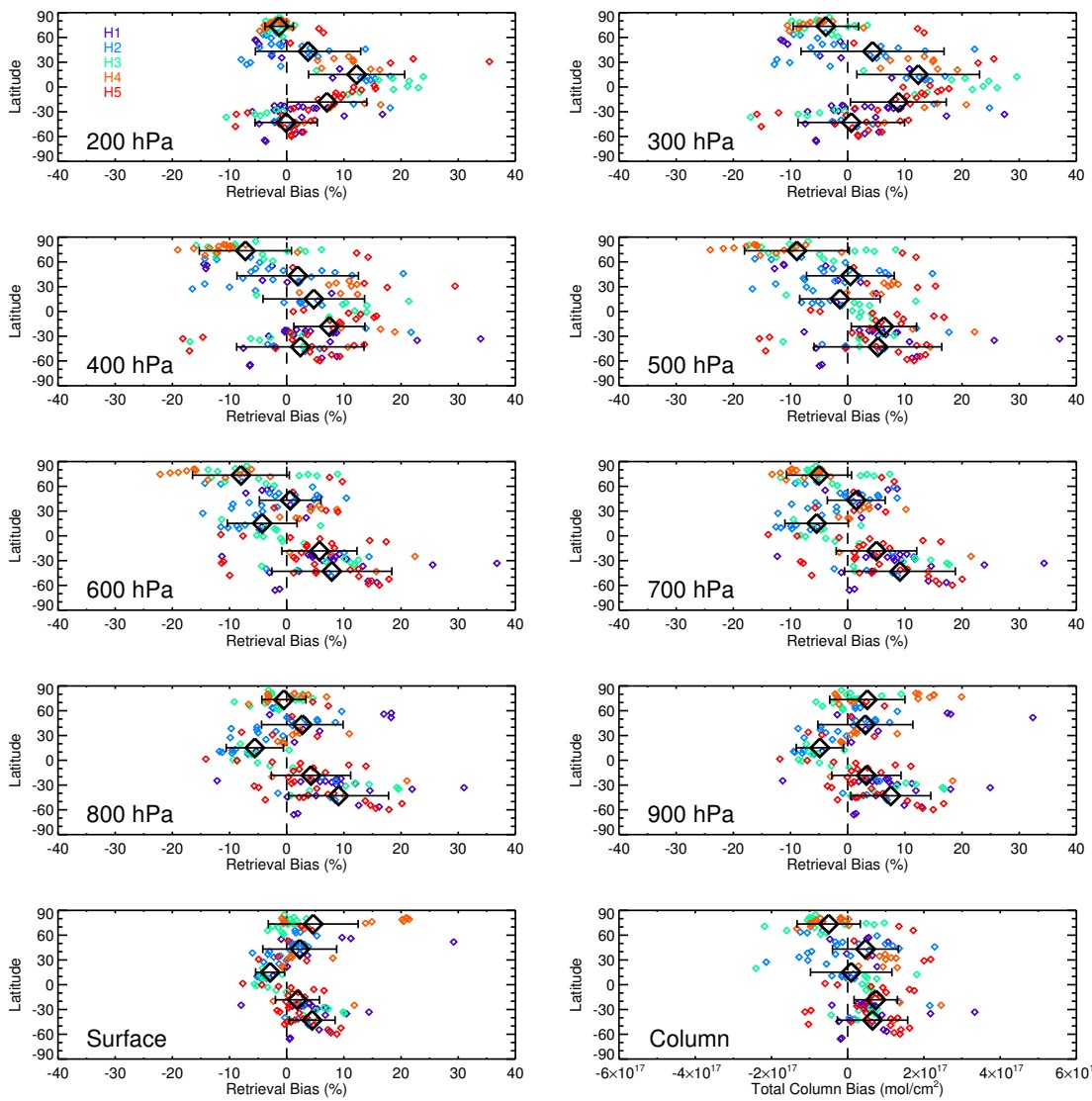

**Figure 2.** Latitude dependence of V7 TIR-only biases based on the HIPPO CO profiles. Results from each of the five stages of HIPPO are color-coded, as indicated by key in top-left panel. Large black diamonds and error bars in each panel indicate bias statistics (mean and standard deviation) representing each 30 degree-wide latitudinal zone.

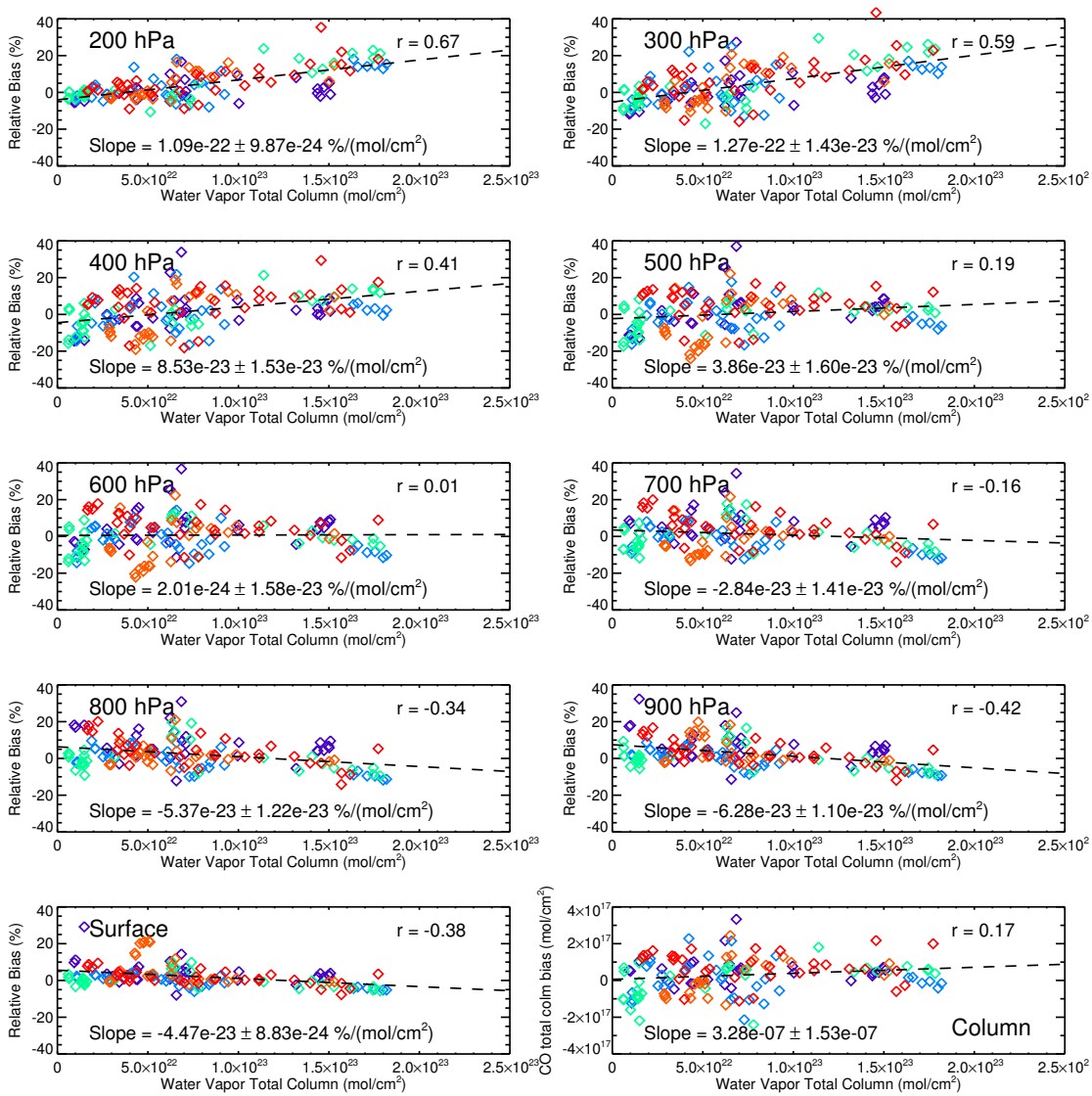

**Figure 3.** Dependence of V7 TIR-only biases on water vapor total column, based on the HIPPO CO profiles. Colors indicate the particular phase of the HIPPO mission, as described in the caption to Figure 2.

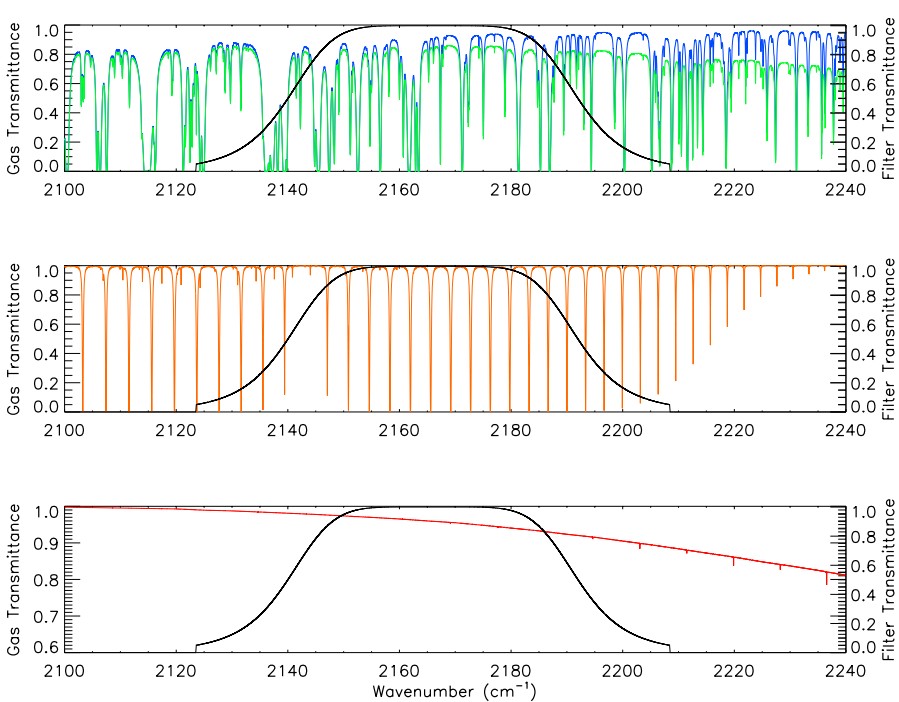

**Figure 4.** Top-of-atmosphere spectral transmittances for water vapor (top panel), CO (middle panel) and nitrogen collisionally-induced absorption (bottom panel) near the MOPITT TIR passband. Top panel compares differences in water vapor absorption calculated with MT_CKD version 1.0 (in blue), used in previous MOPITT forward models, and MT_CKD version 3.2 (in green), used for V8. Bottom panel indicates collisionally-induced absorption by nitrogen, which is represented in V8 forward modeling for the first time. The MOPITT TIR passband is indicated in black in all three panels.

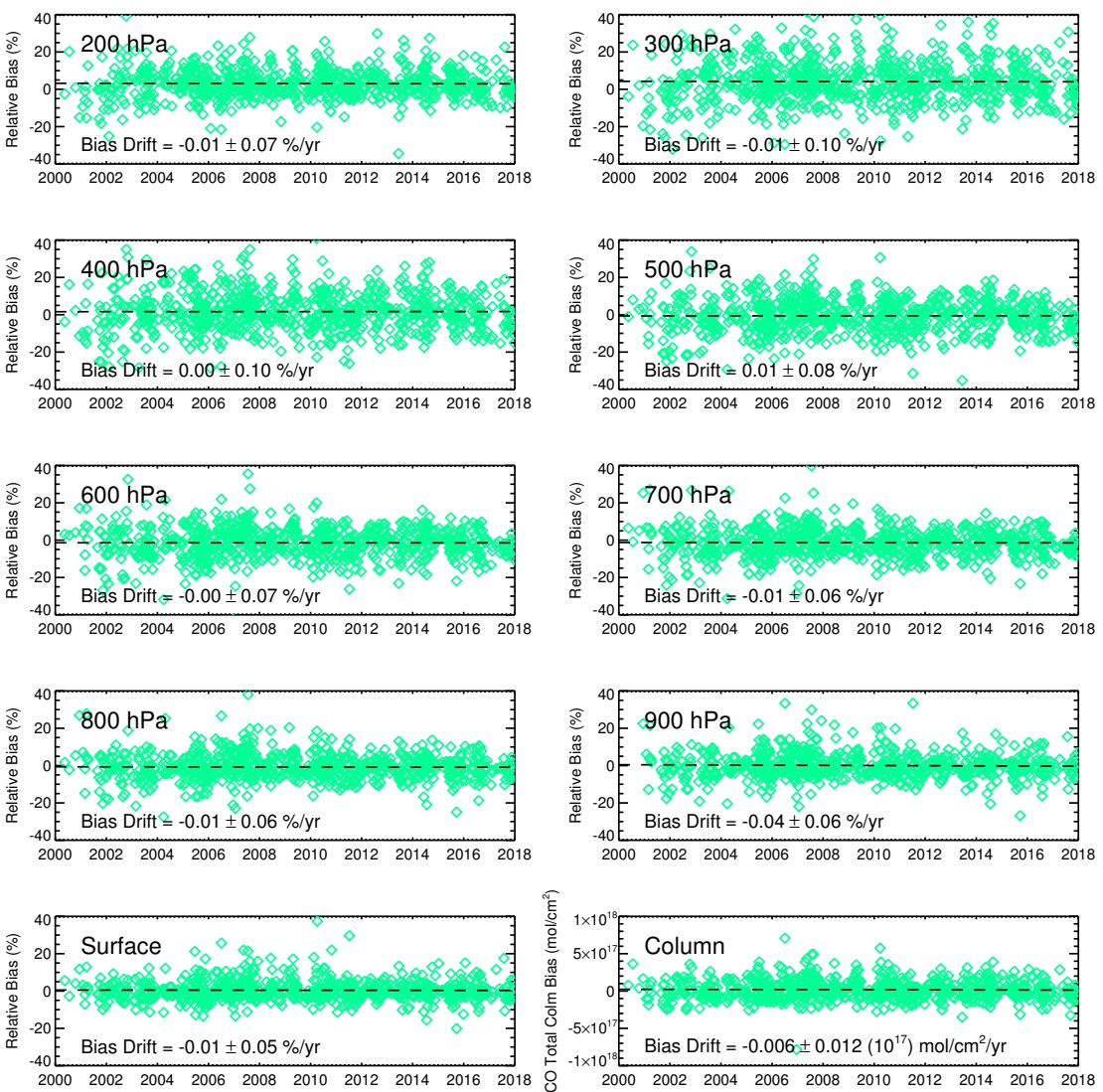

**Figure 5.** Retrieval bias drift for V8 TIR-only products based on the NOAA flask measurements.

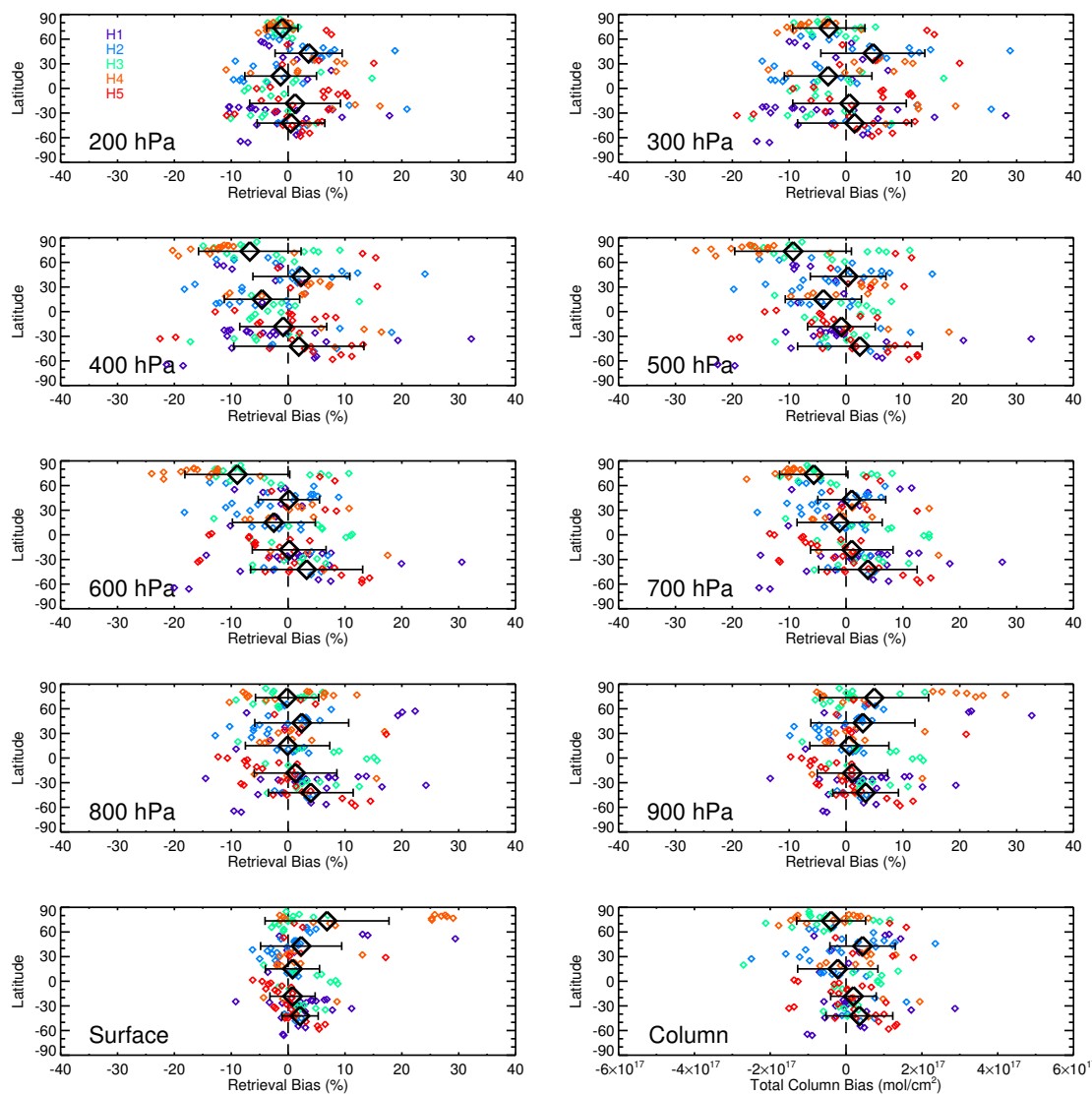

**Figure 6.** Latitude dependence of V8 TIR-only biases (expressed in percent) based on the HIPPO CO profiles. See caption to Fig. 2.

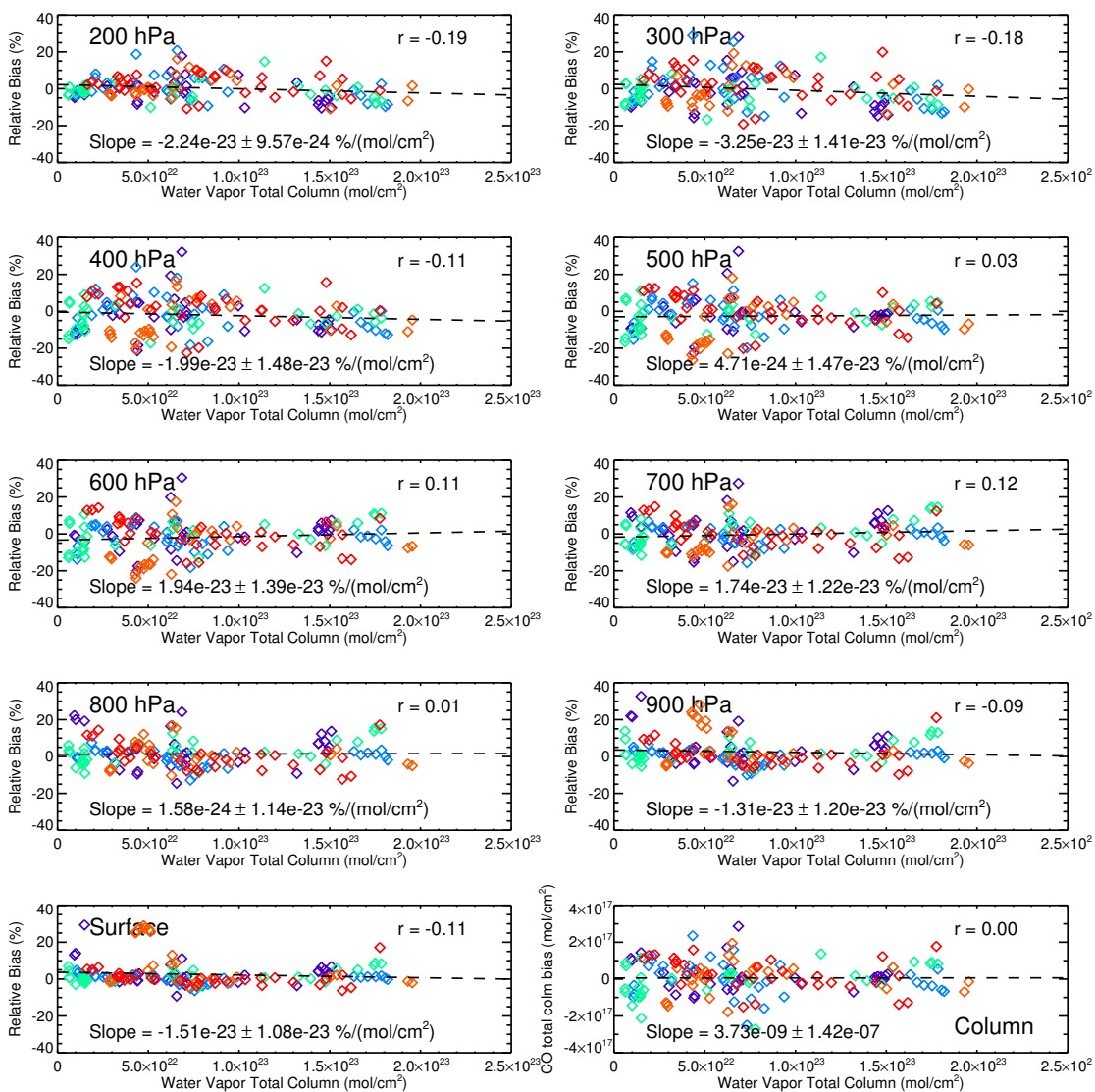

**Figure 7.** Dependence of V8 TIR-only biases on water vapor total column, based on the HIPPO CO profiles. Colors indicate the particular phase of the HIPPO mission, as described in the caption to Figure 2.

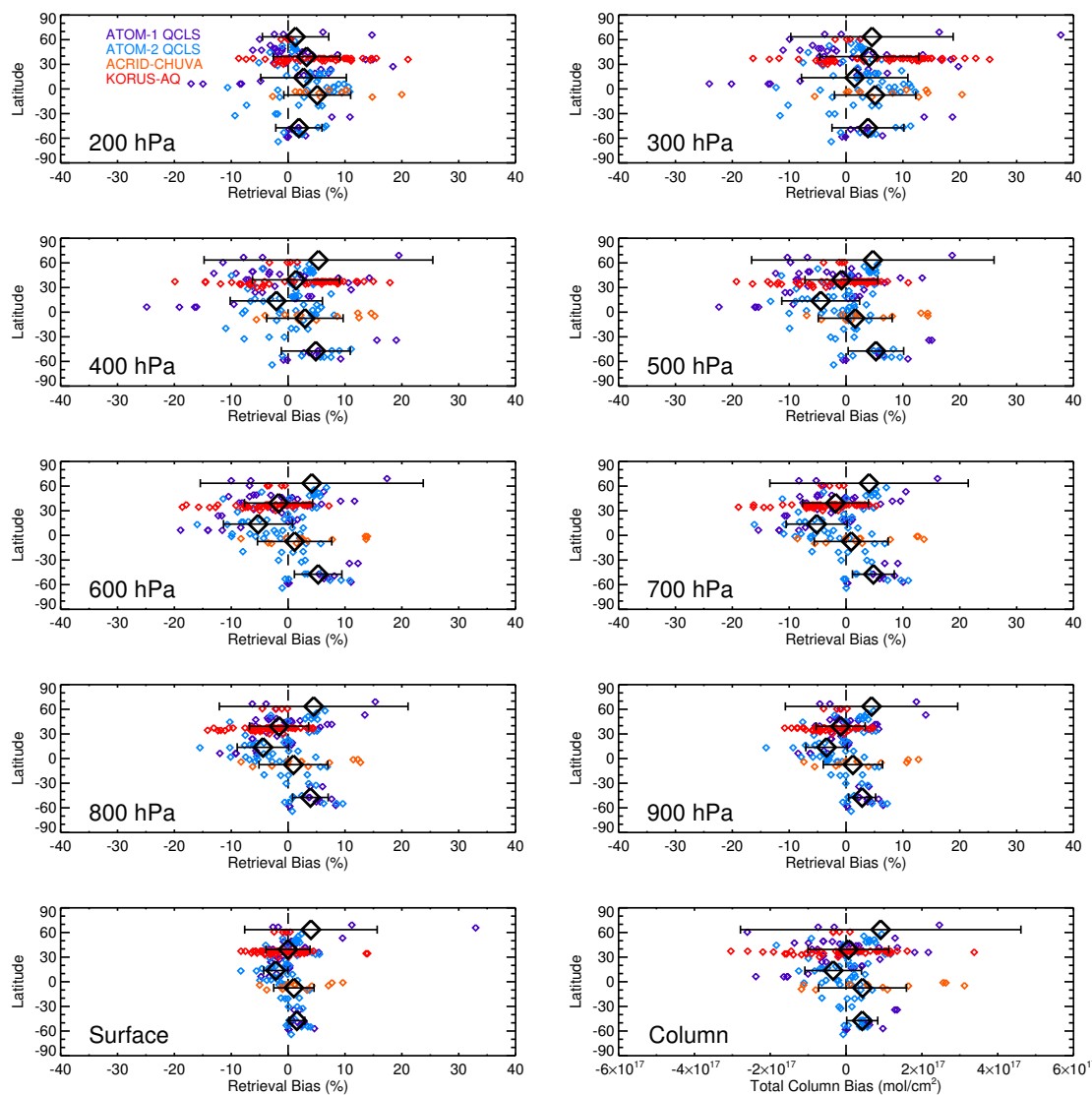

**Figure 8.** V8 TIR-only biases based on the ATom (phases 1 and 2), ACRIDICON, and KORUS-AQ CO profiles. Results are color-coded as indicated by key in top-left panel.

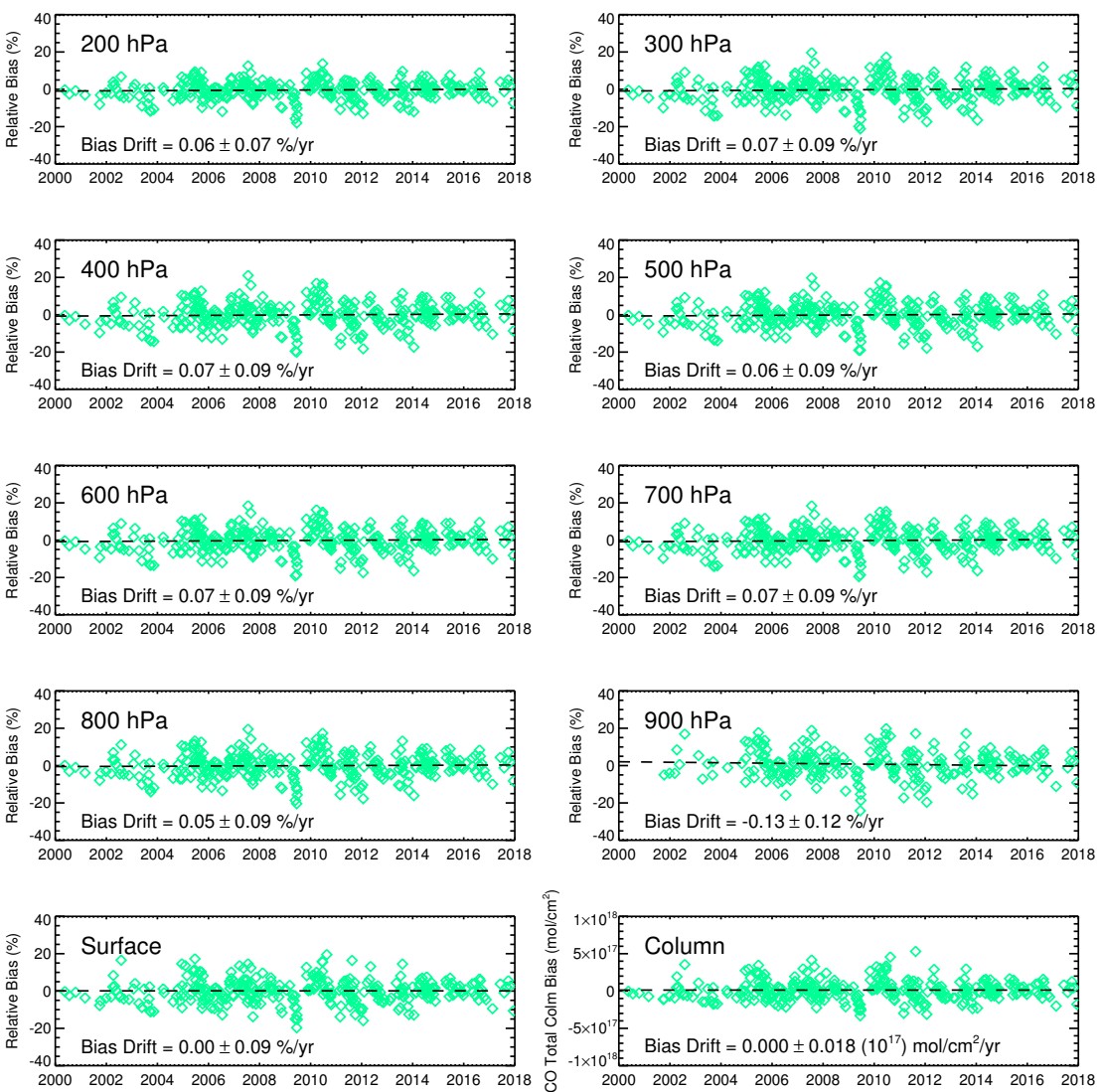

**Figure 9.** Retrieval bias drift for V8 NIR-only products based on the NOAA flask measurements.

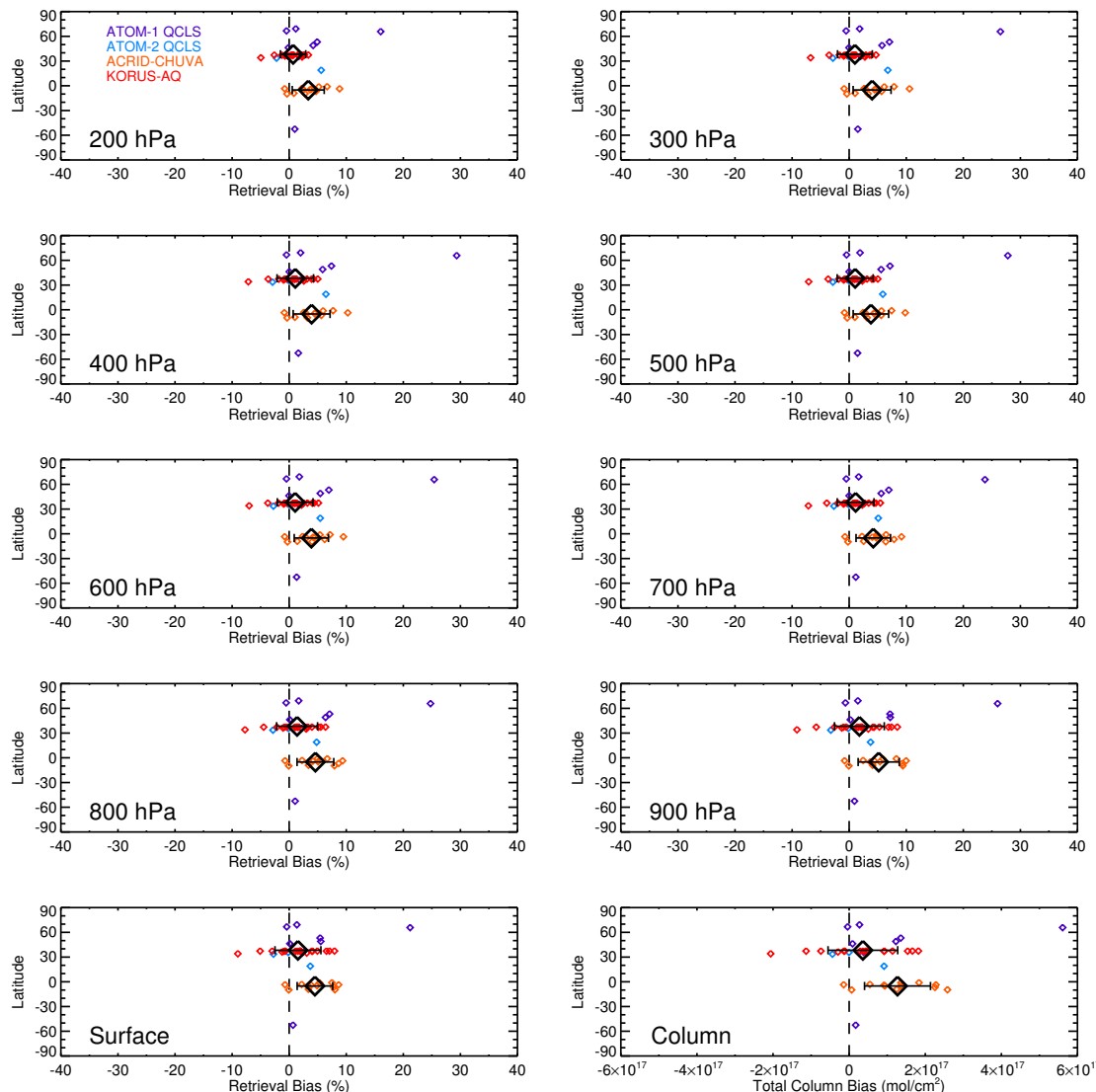

**Figure 10.** V8 NIR-only biases based on the ATom, ACRIDICON, and KORUS-AQ CO profiles.

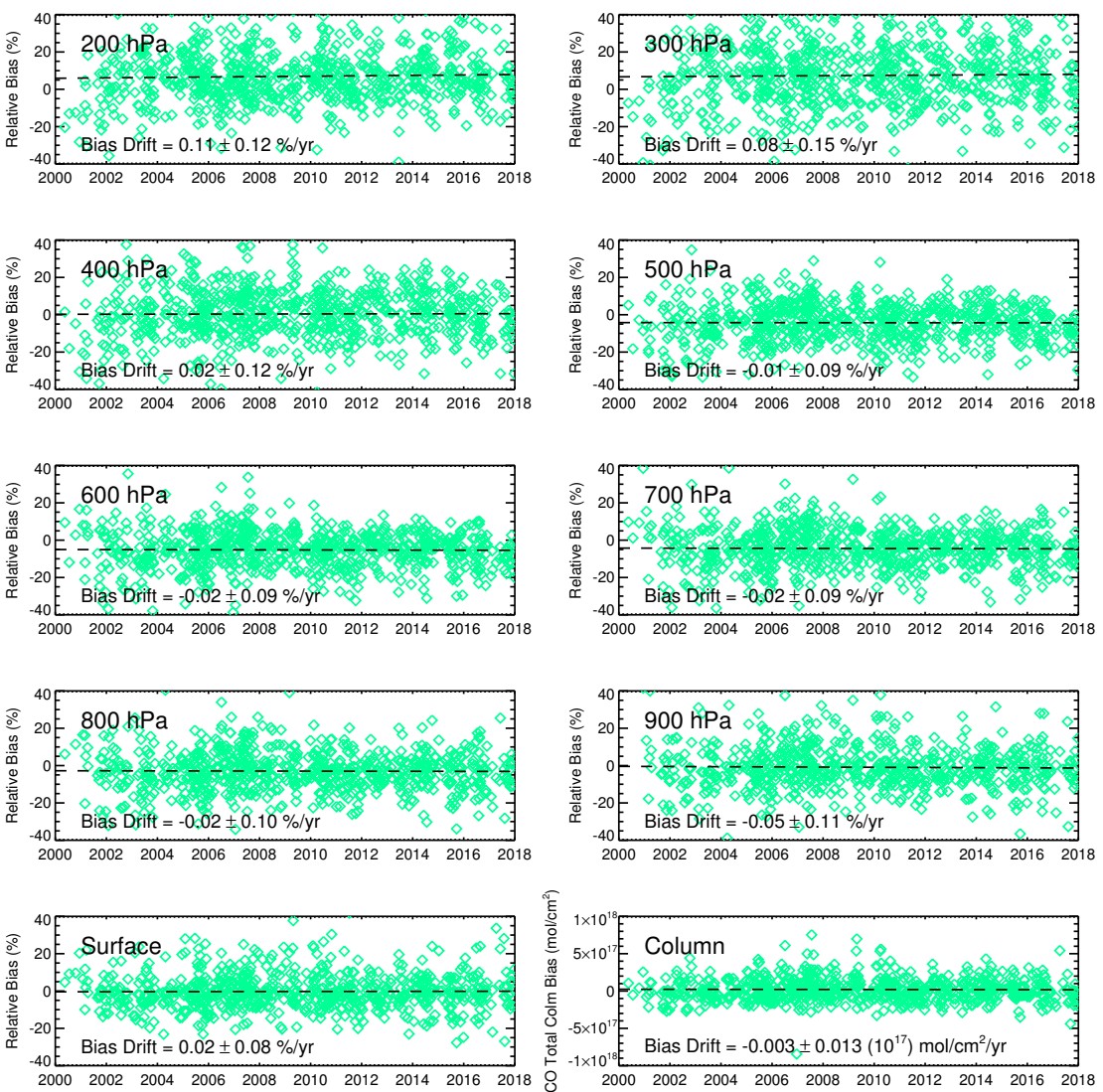

**Figure 11.** Retrieval bias drift for V8 TIR-NIR products based on the NOAA flask measurements.

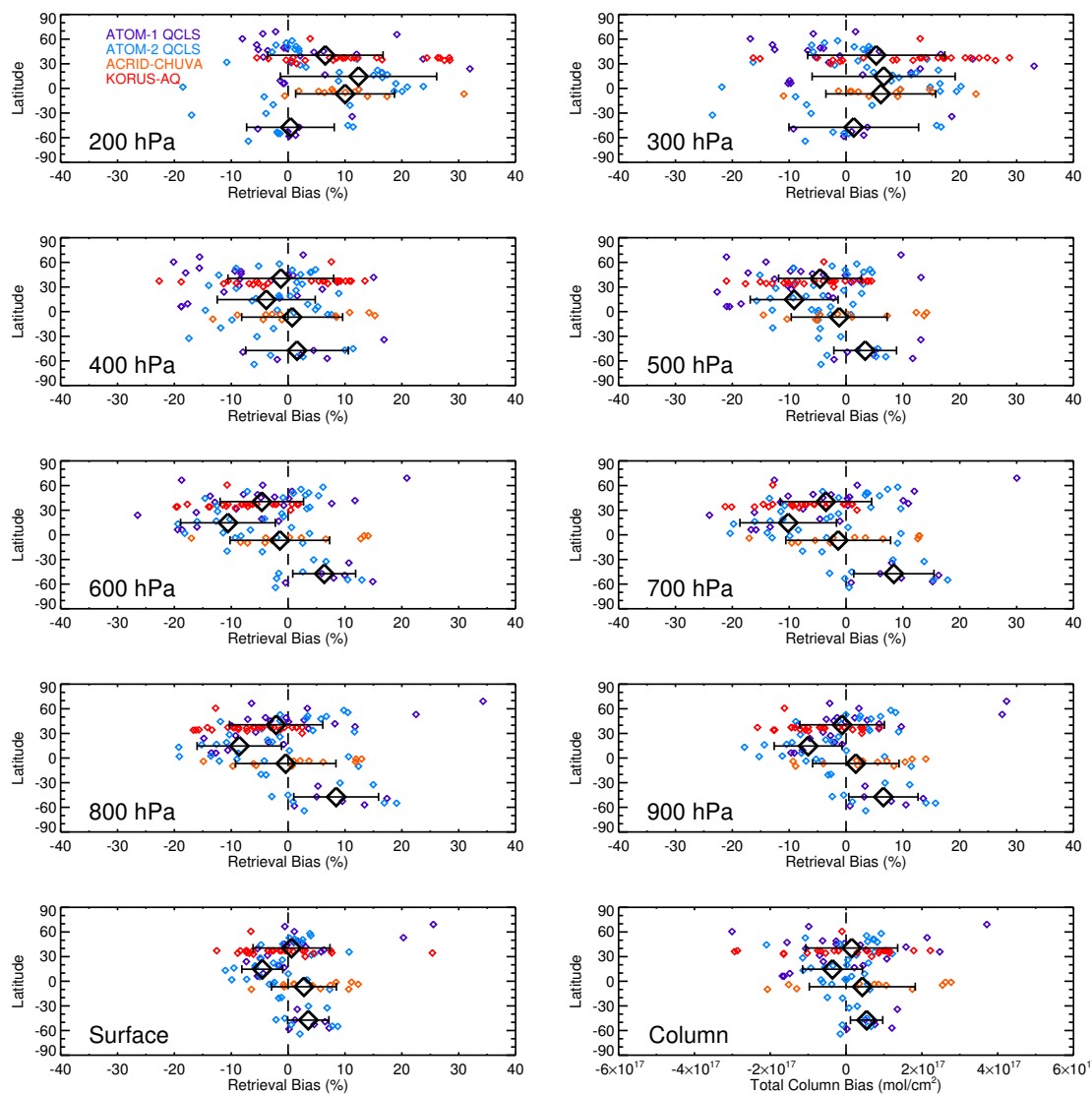

**Figure 12.** V8 TIR-NIR biases based on the ATom, ACRIDICON, and KORUS-AQ CO profiles.

**Table 1.** Radiance bias correction parameters used for processing MOPITT Version 8 retrieval products. See Section 2. $R_0$ is dimensionless. Units of $R_t$ and $R_w$ are $day^{-1}$ and $(molec/cm^2)^{-1}$, respectively.

| | 5A | 5D | 6A | 6D | 7D |
|---|---|---|---|---|---|
| $R_0$ | 1.05970 | 1.04522 | 1.00000 | 0.99522 | 1.04959 |
| $R_t$ | 0.0 | 0.0 | 0.0 | $9.6 \times 10^{-7}$ | $-1.18 \times 10^{-5}$ |
| $R_w$ | 0.0 | $-8.09 \times 10^{-27}$ | 0.0 | 0.0 | $-6.00 \times 10^{-25}$ |

**Table 2.** Characteristics of CO in-situ datasets used for algorithm development (bias minimization) and retrieval validation. See Section 3.1. Number of profiles refers only to aircraft-based profiles which were successfully collocated with MOPITT retrievals.

|  | Observational Period | Region | N Profiles | Technique | Primary Use |
|---|---|---|---|---|---|
| NOAA | 2000-2018 | N. America | 1339 | Flask Samples | Bias Min. |
| HIPPO | 2009-2011 | Pacific Ocean | 358 | QCLS | Bias Min. |
| ACRIDICON-CHUVA | 2014 | Amazonia | 18 | VUV Res. Fluor. | Validation |
| KORUS-AQ | 2016 | S. Korea | 42 | DACOM | Validation |
| ATom | 2016-2017 | Pacific and Atlantic Oceans | 254 | QCLS | Validation |

**Table 3.** Summarized validation results for V7 and V8 TIR-only (V7T and V8T), NIR-only (V7N and V8N) and TIR-NIR (V7J and V8J) products based on in-situ data from NOAA aircraft validation sites. Bias and standard deviation (SD) statistics for the total column are in units of $10^{17}$ molec/cm$^2$. Bias and SD for retrieval levels are expressed in %. Total column drift values are provided both in units of $10^{17}$ molec/cm$^2$/yr and %/yr (in parentheses). Drift for the retrieval levels is expressed in %/yr.

| | | Total Column | Surface | 800hPa | 600hPa | 400hPa | 200hPa |
|---|---|---|---|---|---|---|---|
| V7T | Bias | 0.6 | 1.1 | 0.5 | -0.1 | 1.9 | 3.2 |
| | SD | 2.3 | 6.4 | 7.6 | 8.0 | 11.3 | 9.2 |
| | r | 0.58 | 0.71 | 0.79 | 0.82 | 0.68 | 0.36 |
| | Drift | -0.054 ± 0.020 (-0.767 ± 0.345) | -0.29 ± 0.06 | -0.41 ± 0.06 | -0.18 ± 0.07 | 0.51 ± 0.10 | 0.60 ± 0.08 |
| V8T | Bias | 0.2 | 0.5 | -0.7 | -1.3 | 1.6 | 3.0 |
| | SD | 1.4 | 5.7 | 7.2 | 8.3 | 11.2 | 8.3 |
| | r | 0.82 | 0.74 | 0.77 | 0.80 | 0.72 | 0.54 |
| | Drift | -0.006 ± 0.012 (-0.015 ± 0.061) | -0.01 ± 0.05 | -0.01 ± 0.06 | -0.00 ± 0.07 | 0.00 ± 0.10 | -0.01 ± 0.07 |
| V7N | Bias | -0.1 | -2.1 | -2.4 | -2.4 | -2.5 | -2.2 |
| | SD | 2.4 | 6.4 | 6.7 | 6.4 | 6.8 | 5.1 |
| | r | 0.04 | 0.61 | 0.62 | 0.64 | 0.64 | 0.60 |
| | Drift | -0.149 ± 0.032 (-1.069 ± 0.259) | -0.28 ± 0.09 | -0.27 ± 0.09 | -0.24 ± 0.09 | -0.26 ± 0.09 | -0.20 ± 0.07 |
| V8N | Bias | 0.1 | 0.1 | -0.1 | -0.2 | -0.1 | -0.4 |
| | SD | 1.3 | 6.3 | 6.5 | 6.2 | 6.6 | 4.8 |
| | r | 0.60 | 0.60 | 0.62 | 0.64 | 0.64 | 0.61 |
| | Drift | 0.000 ± 0.018 (0.049 ± 0.107) | 0.00 ± 0.09 | 0.05 ± 0.09 | 0.07 ± 0.09 | 0.07 ± 0.09 | 0.06 ± 0.07 |
| V7J | Bias | 0.6 | 0.8 | -0.5 | -3.2 | -0.7 | 5.1 |
| | SD | 2.6 | 11.2 | 12.8 | 10.1 | 14.6 | 17.0 |
| | r | 0.57 | 0.59 | 0.70 | 0.80 | 0.56 | 0.11 |
| | Drift | -0.076 ± 0.022 (-1.082 ± 1.780) | -0.67 ± 0.09 | -0.94 ± 0.10 | -0.49 ± 0.09 | 0.77 ± 0.12 | 1.29 ± 0.14 |
| V8J | Bias | 0.2 | -0.1 | -2.7 | -5.1 | 0.2 | 6.7 |
| | SD | 1.6 | 9.8 | 11.7 | 10.6 | 14.1 | 14.7 |
| | r | 0.81 | 0.62 | 0.68 | 0.76 | 0.64 | 0.30 |
| | Drift | -0.003 ± 0.013 (0.001 ± 0.070) | 0.02 ± 0.08 | -0.02 ± 0.10 | -0.02 ± 0.09 | 0.02 ± 0.12 | 0.11 ± 0.12 |

**Table 4.** Latitude dependence of validation results for V7 and V8 TIR-only products as indicated using in-situ data from HIPPO field campaign, and corresponding to results shown in Figures 2 and 6. See caption to Table 2.

| | | | Total Column | Surface | 800hPa | 600hPa | 400hPa | 200hPa |
|---|---|---|---|---|---|---|---|---|
| V7T | 60N:90N | Bias | -0.50 | 4.6 | -0.5 | -8.0 | -7.2 | -1.3 |
| | | SD | 0.83 | 7.9 | 3.9 | 8.5 | 8.1 | 2.5 |
| | 30N:60N | Bias | 0.47 | 2.3 | 2.7 | 0.6 | 1.9 | 3.7 |
| | | SD | 0.87 | 6.5 | 7.1 | 5.4 | 10.6 | 9.2 |
| | Eq:30N | Bias | 0.09 | -2.9 | -5.6 | -4.3 | 4.7 | 12.2 |
| | | SD | 1.07 | 2.5 | 5.0 | 6.1 | 8.9 | 8.4 |
| | 30S:Eq | Bias | 0.74 | 1.9 | 4.3 | 5.7 | 7.5 | 7.0 |
| | | SD | 0.57 | 3.8 | 6.9 | 6.6 | 6.2 | 7.0 |
| | 60S:30S | Bias | 0.65 | 4.5 | 9.1 | 7.9 | 2.4 | -0.1 |
| | | SD | 0.92 | 4.0 | 8.7 | 10.5 | 11.1 | 5.4 |
| V8T | 60N:90N | Bias | -0.39 | 6.8 | -0.2 | -8.9 | -6.7 | -1.0 |
| | | SD | 0.91 | 10.9 | 5.6 | 9.2 | 9.0 | 2.8 |
| | 30N:60N | Bias | 0.44 | 2.3 | 2.4 | 0.1 | 2.3 | 3.6 |
| | | SD | 0.86 | 7.1 | 8.3 | 5.4 | 8.5 | 5.9 |
| | Eq:30N | Bias | -0.22 | 0.8 | -0.1 | -2.5 | -4.6 | -1.3 |
| | | SD | 1.06 | 4.8 | 7.4 | 7.3 | 6.6 | 6.3 |
| | 30S:Eq | Bias | 0.20 | 0.8 | 1.3 | 0.2 | -0.9 | 1.2 |
| | | SD | 0.60 | 4.0 | 7.3 | 6.5 | 7.6 | 8.0 |
| | 60S:30S | Bias | 0.35 | 2.1 | 4.0 | 3.3 | 1.9 | 0.5 |
| | | SD | 0.88 | 3.2 | 7.5 | 9.9 | 11.5 | 6.0 |

**Table 5.** Summarized validation results for V8 products based on in-situ data from ACRIDICON-CHUVA, KORUS-AQ, and ATom field campaigns. See caption to Table 2.

|          |     |      | Total Column | Surface | 800hPa | 600hPa | 400hPa | 200hPa |
|----------|-----|------|--------------|---------|--------|--------|--------|--------|
| ACRIDICON | V8T | Bias | 0.8 | 1.9 | 2.9 | 3.6 | 5.3 | 6.6 |
|          |     | SD   | 1.5 | 4.6 | 7.0 | 7.3 | 7.2 | 6.3 |
|          |     | r    | 0.79 | 0.82 | 0.82 | 0.77 | 0.75 | 0.80 |
|          | V8N | Bias | 1.3 | 4.5 | 4.6 | 3.9 | 3.9 | 3.3 |
|          |     | SD   | 0.9 | 3.1 | 3.2 | 3.0 | 3.2 | 2.8 |
|          |     | r    | 0.82 | 0.83 | 0.83 | 0.81 | 0.80 | 0.80 |
|          | V8J | Bias | 0.7 | 4.5 | 1.2 | 0.5 | 1.8 | 10.4 |
|          |     | SD   | 1.6 | 5.8 | 8.9 | 9.7 | 9.3 | 8.2 |
|          |     | r    | 0.80 | 0.88 | 0.82 | 0.65 | 0.65 | 0.87 |
| KORUS-AQ | V8T | Bias | 0.0 | -0.8 | -2.9 | -2.5 | 2.0 | 4.2 |
|          |     | SD   | 1.1 | 4.5 | 4.5 | 5.4 | 7.5 | 5.5 |
|          |     | r    | 0.91 | 0.84 | 0.90 | 0.86 | 0.79 | 0.73 |
|          | V8N | Bias | 0.3 | 1.5 | 1.1 | 0.8 | 0.8 | 0.5 |
|          |     | SD   | 1.0 | 4.3 | 3.6 | 3.0 | 3.0 | 2.0 |
|          |     | r    | 0.67 | 0.69 | 0.66 | 0.64 | 0.64 | 0.66 |
|          | V8J | Bias | 0.1 | -1.5 | -6.6 | -7.5 | 0.8 | 11.2 |
|          |     | SD   | 1.3 | 7.5 | 6.0 | 6.3 | 10.0 | 10.3 |
|          |     | r    | 0.89 | 0.77 | 0.83 | 0.81 | 0.77 | 0.68 |
| ATom     | V8T | Bias | 0.2 | 0.8 | 0.8 | 0.4 | 1.2 | 1.9 |
|          |     | SD   | 1.5 | 5.0 | 7.9 | 9.4 | 9.8 | 5.8 |
|          |     | r    | 0.70 | 0.47 | 0.67 | 0.77 | 0.81 | 0.77 |