# Peer review of "Radiance-based Retrieval Bias Mitigation for the MOPITT Instrument: The Version 8 Product"

_Atmospheric Measurement Techniques, 2019_

## Referee Comment (RC1) · Anonymous Referee #1 · 22 Mar 2019

This work is a useful description of the improvements introduced in the latest version 8 of CO data derived from measurements recorded with the MOPITT instrument onboard Terra. The empirical foundation of the detected drifts and implemented countermeasures is the comparison of MOPITT data with airborne in situ observations of CO collected regularly by NOAA and during several campaigns. The paper is well written, in some parts perhaps slightly too concise (see minor comment at the end of this review) and the statistical analysis is carried out in a careful manner. I recommend publication after minor revisions.

I have only two reservations concerning the presented work worth mentioning:

[Figure]

(1) It would be a valuable independent confirmation of the significant reduction of drifts claimed for version 8 to incorporate the rich dataset of CO provided by ground-based FTIR networks TCCON and NDACC, as these observations also cover the complete lifetime of the satellite. The TCCON data would be useful for checking the spurious trends of CO total column in version 7 and version 8, the NDACC observations even offer a moderate vertical resolution. I agree that one will assume that the in-situ dataset is the primary reference (as also used for calibrating ground-based remote sensing observations), but the remote sensing network observations cover additional locations and times. Moreover, by the inherent character of a spectroscopic solar absorption measurement performed by a distributed network, a common drift of the network as a whole is highly unlikely, so these datasets seem a useful asset especially in the context of trend evaluation.

(2) For the non-experts (including myself) with just a schematic understanding of the working principle underlying the MOPITT instrument, the idea of the proposed radiance-based correction remains somewhat obscure. The explanation provided "potential bias sources including errors in instrumental specifications, forward model error. . ., errors in assumed spectroscopic data, and geophysical errors" is focusing mainly on aspects of the data analysis and the spectral simulations required in this context (as does the provided reference Deeter et al., 2014). I understand that model errors will evoke retrieval errors, however, as a steady trend is observed this seems to be an instrument issue? Could you make a clearer statement in this respect? Might it be even possible to provide further speculations concerning what is actually drifting here, detectors, sources, window transmissions of gas cells, . . .? Although the improvements in the version 8 data are obvious, such kind of intervention by incorporating an explicit time-dependent correction is to some degree problematic. The previous publication on version 7 data was entitled "a climate-scale satellite record for carbon monoxide". Unfortunately, as version 8 introduces an ad-hoc time-dependent correction, the confession shines through that MOPITT is not capable of measuring CO trends with higher confidence than 0.5 to 1% / a on a decadal scale, as an ad-hoc

correction is needed for adjusting the results (so the trend on decadal scales is actually determined by the aircraft data, not MOPITT). If some sort of instrumental degradation could be identified and quantified independently from atmospheric observations using routine onboard calibration measurements (I do not know the procedures – observations of internal blackbody sources or a solar diffuser?) and finally be made responsible using a quantitative error-propagating scheme built on an instrument model, this point of criticism could be removed. It might be appropriate to include a deeper discussion of these issues.

Minor comment: Facing this multitude of validation datasets is a bit confusing for the reader. Although all activities and campaigns are cited appropriately, it would be useful to provide some kind of tabulated or graphical overview in order to summarize the temporal (and perhaps spatial?) coverage of each dataset. This would also incorporate useful information on the relation of the subset of AC, KORUS-AQ and Atom with the datasets used for the development of the bias minimization. Finally, in the several tables reporting bias values, SD, r, (and drifts in case of table 2) . . . it would be useful to also specify how many aircraft profiles were used for calculating these statistical parameters.
* * *

---

## Referee Comment (RC2) · Anonymous Referee #2 · 25 Mar 2019

The authors provide an interesting, comprehensive, and well-written account of the latest MOPITT CO retrieval update (V8). This article can be published after minor revisions, which ideally address the following two points at least:

(a) Page 4 lines 11-12 mentions that the "Values of these parameters [...] were obtained by minimizing retrieval biases for the TIR-only validation results." The last paragraph of Section 3.1 again mentions the methodology in terms of the NOAA and HIPPO 'validation' for bias minimization, versus the campaign validation for verification of the minimization method. This type of phrasing (terminology) and text structure is very confusing and to some extent incorrect: The TIR-only comparisons are a validation for

[Figure]

V7 only, while they are merely a bias minimization methodology check for V8. One cannot call the latter a validation any longer. It is therefore recommended to move the content of Section 3.2 to Section 2.2 (or clearly restructure differently), and adopt the terminology from 'validation' to comparative 'verification' accordingly. This shift would also bring more focus on the bias mitigation that is mentioned in the manuscript title.

(b) Section 3: It would be very helpful to provide a map of the in-situ reference data flights with an indication of where collocations have been used for data comparison. It should become very clear than where the validation results are valid, and which data have been used for bias mitigation only.

Minor / technical remarks: - The introduction is quite long. The general description of the MOPITT retrieval process could go into a separate section. - Page 1, line 10-11: "MOPITT [. . .] which produces retrievals" does not sound correct. - Page 1, line 21: "the ultimate CO product" rather sounds either conceited or fatalist. Something more neutral might have been the intention. - Page 2, second paragraph: Please mention the TIR and NIR wavelength ranges. - Page 2: Figure numbering in the text jumps from 1 to 3; Figure 2 is not mentioned. - Section 2: Please briefly introduce the four aspects of the retrieval enhancement at the beginning of this section. - Page 5 lines 13-17 are not fully clear. Why change the relative channel 7 average radiance in order to obtain consistency between V7 and V8 processing? - Section 3.1: Please briefly elaborate on the "flask sampling system" - Page 7 lines 1-6 are not fully clear. First it is said that the in-situ measurements are vertically extended, than it is said that validation results for the 100 hPa level are not reported because of a lack of in-situ measurements above this level. - Page 8 lines 28-29: Does the last sentence of Section 3.4 indicate that all significant bias has been removed and that only 5 % spread remains? This is not fully clear from the current phrasing. - Figures: The authors should explain in the text how the "bias drift", "slope", and the uncertainties thereon are calculated.

---

## Referee Comment (RC3) · Anonymous Referee #4 · 28 Mar 2019

This paper describes version8 of the MOPITT CO retrievals and of its associated data record. Despite being short and technical, the paper deals with an increasingly important aspect of satellite remote sensing, which is the consistency and homogeneity in the time series at level 2. By targeting CO, a so-called indirect climate variable, and relying on the longest time-series available from satellites (almost two decades with MOPITT), this paper is a very useful reference. To achieve this consistency in the CO time series for version 8, an empirical and "dynamical" radiance correction scheme is proposed. The methodology is well described and the results give convincing evidence of its benefit in V8 over previous versions. New validation results using recent aircraft measurement campaigns (from ACRIDICON-CHUVA, KORUS-AQ and Atom)

[Figure]

are given as well, and these provide an additional support for the proposed radiance-bias minimization scheme. The paper is extremely clear and very well written.

For these reasons, I recommend publication of the paper in AMT. I list below a few comments or questions that the authors may want to consider to possibly improve or clarify some aspects of the paper.

- Page2, discussion around Figure 1: the drifts compared to the measurement flasks are mostly positive below 500 hPA and negative above. In the total columns this results in almost no drift. This could be indicated from the start (around line 25).

- Page 2, line 30: To explain the latitudinal-dependent bias in the V7-TIR only CO, the water vapor field is suspected to be a cause. From Figure 3 it seems indeed obvious that water plays a role and this is one of the reason for the bias correction » suggestion to remove "perhaps" on line 30?

- Would a figure showing the (global? hemispheric?) averaged retrieved profile from v7 and V8 with the associated standard deviation not be insightful? Is this possible and does this add something?

- Page 5 top; it would be useful to see how to new way to calculate the averaging kernels in v8 affects the averaging kernel calculations. Also, is there an impact on the validation between V7 and V8 that could potentially be related to the AVKs calculations?

- Page 5, line 10. Remove parentheses for that sentence?

- Page 5, equation 5. Is x_true different than x_cmp from eq. (3)? I understand that x_cmp is more general but for this paper is the distinction useful?

- Page 6, last sentence before section 3.1; I would suggest to move this sentence earlier in section 2.3

- Figure 5 is similar to Figure 1 for the new version. When discussing Figure 5 on page

7 it would be good to refer the reader to Figure 1 for a useful comparison

- Figures 5 and 6 are convincing on what the dynamic bias-correction brings. Nice results; well done.

- Figure 8 compares the new CO profiles from V8 to more recent reference measurements from aircrafts. As said before, this is useful and again convincing. It would, however, be nice to show (or at least comment) how V7T performs compared to these reference dataset (in comparison to V8T).

- Page 8. The last sentence before section 3.3 refers to biases at high northern latitude but looking at Figure 8 one has the feeling that there are almost no measurements in the latitude band 60-90 N. Is that the case and could you please then indicate how significant this bias is?

- Comparing Figure 12 and Fig8 one sees that the biases in v8 are larger for the TIR-NIR retrievals than for the TIR only. Was this feature also there for v7? Is there a known reason for this?

- Check if the reference Worden et al., 2013 is cited in the text.

- Caption of table 2: correlation coefficients are given between retrieved quantities and corresponding a priori quantities. Is it a priori or reference?

- Would it not be possible to make a single table out if Tables 4-6? This would be nice and also helpful to easily compare the results from each validation set.

---

## Author Comment (AC1) · 24 May 2019

Authors' Responses to Reviewer #1 of 'Radiance-based Retrieval Bias Mitigation for the MOPITT Instrument: The Version 8 Product' by M. Deeter et al.

Original reviewer's comments in blue. Authors' responses in black.

**Replies to Comments of Reviewer #1**

This work is a useful description of the improvements introduced in the latest version 8 of CO data derived from measurements recorded with the MOPITT instrument onboard Terra. The empirical foundation of the detected drifts and implemented countermeasures is the comparison of MOPITT data with airborne in situ observations of CO collected regularly by NOAA and during several campaigns. The paper is well written, in some parts perhaps slightly too concise (see minor comment at the end of this review) and the statistical analysis is carried out in a careful manner. I recommend publication after minor revisions. I have only two reservations concerning the presented work worth mentioning:

(1) It would be a valuable independent confirmation of the significant reduction of drifts claimed for version 8 to incorporate the rich dataset of CO provided by ground-based FTIR networks TCCON and NDACC, as these observations also cover the complete lifetime of the satellite. The TCCON data would be useful for checking the spurious trends of CO total column in version 7 and version 8, the NDACC observations even offer a moderate vertical resolution. I agree that one will assume that the in-situ dataset is the primary reference (as also used for calibrating ground-based remote sensing observations), but the remote sensing network observations cover additional locations and times. Moreover, by the inherent character of a spectroscopic solar absorption measurement performed by a distributed network, a common drift of the network as a whole is highly unlikely, so these datasets seem a useful asset especially in the context of trend evaluation.

Response: We agree that comparisons between MOPITT and TCCON/NDACC could be useful for validation of trends, but we consider comparisons against in-situ profiles more definitive. Significant differences in column averaging kernels for MOPITT and TCCON/NDACC retrievals complicate the comparisons considerably and inevitably raise questions related to the possible influence of long-term changes in CO concentrations on apparent biases (and drift) between MOPITT and TCCON/NDACC. The use of in-situ profiles for validation avoids this issue entirely. Ongoing MOPITT validation work includes comparisons with TCCON (Hedelius et al., in prep) and NDACC (Buchholz et al., AMT, 2017 – to be updated for V8) as well as other long-term satellite CO datasets (e.g., IASI).

(2) For the non-experts (including myself) with just a schematic understanding of the working principle underlying the MOPITT instrument, the idea of the proposed radiance-based correction remains somewhat obscure. The explanation provided "potential bias sources including errors in instrumental specifications, forward model error. . ., errors in assumed spectroscopic data, and geophysical errors" is focusing mainly on aspects of the data analysis and the spectral simulations required in this context (as does the provided reference Deeter et al., 2014). I understand that model errors will evoke retrieval errors, however, as a steady trend is observed this seems to be an instrument issue? Could you make a clearer statement in this respect?

Response: We refer to Drummond et al., 2010 for more details on MOPITT measurement concepts. Possible sources of the bias drift are briefly discussed in the second paragraph of the Conclusion. While an instrumental source is quite possible, it is also conceivable that the quality of the MERRA-2 meteorological data (temperature and water vapor profiles) used in the retrieval algorithm has varied

over the MOPITT mission (due to the assimilation of an ever-evolving set of in-situ and satellite datasets). We feel it would be premature to make further statements on this topic.

Might it be even possible to provide further speculations concerning what is actually drifting here, detectors, sources, window transmissions of gas cells, . . .?

Response: Collaborations with the instrument developers will be essential to identify possible physical sources to the bias drift, if in fact the bias source is instrumental. Preliminary discussions with the Canadian MOPITT team (who designed the instrument) have taken place, but no conclusions have yet been reached. We do not feel that it would be appropriate to speculate further on this issue in this manuscript.

Although the improvements in the version 8 data are obvious, such kind of intervention by incorporating an explicit time-dependent correction is to some degree problematic. The previous publication on version 7 data was entitled "a climate-scale satellite record for carbon monoxide". Unfortunately, as version 8 introduces an ad-hoc time-dependent correction, the confession shines through that MOPITT is not capable of measuring CO trends with higher confidence than 0.5 to 1% / a on a decadal scale, as an ad-hoc correction is needed for adjusting the results (so the trend on decadal scales is actually determined by the aircraft data, not MOPITT). If some sort of instrumental degradation could be identified and quantified independently from atmospheric observations using routine onboard calibration measurements (I do not know the procedures – observations of internal blackbody sources or a solar diffuser?) and finally be made responsible using a quantitative error-propagating scheme built on an instrument model, this point of criticism could be removed. It might be appropriate to include a deeper discussion of these issues.

Response: Our chosen strategy to rely on the stability of the NOAA aircraft in-situ data to determine the time-dependent radiance bias correction factors does imply that the fidelity of CO trends in the radiance bias-corrected MOPITT data (i.e., the V8 retrieval product) are ultimately limited by the stability of the NOAA in-situ measurements. However, the NOAA measurements are widely accepted as the standard for long-term CO analyses, and the stability of the NOAA flask-sample measurements are certainly much better than 0.5 to 1% per year. The NOAA dataset is therefore quite suitable for climate analyses. On the other hand, we acknowledge (e.g., in Section 3.2) that bias drift in the V8 product for regions not represented by the NOAA aircraft network (primarily covering North America) could be substantially different. This seems unlikely though, particularly if the source of the bias is instrumental. We are currently investigating methods for quantifying MOPITT bias drift globally, using field campaign measurements as well as both ground-based and satellite-based remote sensing datasets.

Minor comment: Facing this multitude of validation datasets is a bit confusing for the reader. Although all activities and campaigns are cited appropriately, it would be useful to provide some kind of tabulated or graphical overview in order to summarize the temporal (and perhaps spatial?) coverage of each dataset. This would also incorporate useful information on the relation of the subset of AC, KORUS-AQ and Atom with the datasets used for the development of the bias minimization.

Response: A table has been added (introduced in Section 3) in the revised manuscript summarizing these aspects of the different in-situ datasets.

Finally, in the several tables reporting bias values, SD, r, (and drifts in case of table 2) . . . it would be useful to also specify how many aircraft profiles were used for calculating these statistical parameters.

Response: The number of aircraft profiles for each in-situ dataset is included in the new table.

---

## Author Comment (AC2) · 24 May 2019

Authors' Responses to Reviewer #2 of 'Radiance-based Retrieval Bias Mitigation for the MOPITT Instrument: The Version 8 Product' by M. Deeter et al.

Original reviewer's comments in blue.  Authors' responses in black.

**Replies to Comments of Reviewer #2**

The authors provide an interesting, comprehensive, and well-written account of the latest MOPITT CO retrieval update (V8). This article can be published after minor revisions, which ideally address the following two points at least:

(a) Page 4 lines 11-12 mentions that the "Values of these parameters [. . .] were obtained by minimizing retrieval biases for the TIR-only validation results." The last paragraph of Section 3.1 again mentions the methodology in terms of the NOAA and HIPPO 'validation' for bias minimization, versus the campaign validation for verification of the minimization method. This type of phrasing (terminology) and text structure is very confusing and to some extent incorrect: The TIR-only comparisons are a validation for V7 only, while they are merely a bias minimization methodology check for V8. One cannot call the latter a validation any longer. It is therefore recommended to move the content of Section 3.2 to Section 2.2 (or clearly restructure differently), and adopt the terminology from 'validation' to comparative 'verification' accordingly. This shift would also bring more focus on the bias mitigation that is mentioned in the manuscript title.

Response: We agree that it is important to distinguish uses of in-situ datasets for algorithm development (e.g., bias minimization) as opposed to independent validation.  This issue is fully addressed in the last paragraph of Section 3.1 and is mentioned as well in both the Introduction and Conclusion.  Thus, we don't believe readers will be confused about this issue.  The last column in the added table (as suggested by the first Reviewer) will emphasize this point even further.

(b) Section 3: It would be very helpful to provide a map of the in-situ reference data flights with an indication of where collocations have been used for data comparison. It should become very clear than where the validation results are valid, and which data have been used for bias mitigation only.

Response: See the response to the next-to-last comment by Reviewer #1 above.  The new table emphasizes which datasets are used for algorithm development and which for independent validation.

Minor / technical remarks: - The introduction is quite long. The general description of the MOPITT retrieval process could go into a separate section.

Response: The Introduction currently includes five paragraphs, providing an overview of the MOPITT mission, MOPITT products, and validation results for the Version 7 product.  We feel that this section provides the necessary background for the remaining sections of the paper without delving too deeply into the details.

- Page 1, line 10-11: "MOPITT [. . .] which produces retrievals" does not sound correct.

Response: The word 'produces' has been replaced with 'permits'.  The full sentence now reads 'MOPITT ("Measurements of Pollution in the Troposphere") is a gas correlation radiometer instrument on the NASA Terra satellite which permits retrievals of CO vertical profiles using both thermal-infrared (TIR) and near-infrared (NIR) measurements.'

- Page 1, line 21: "the ultimate CO product" rather sounds either conceited or fatalist. Something more neutral might have been the intention.

Response: The phrase 'ultimate CO product' was meant to indicate the product which results from the bias mitigation strategy. 'Resulting CO product' is now used instead.

- Page 2, second paragraph: Please mention the TIR and NIR wavelength ranges.

Response: Agreed. The following sentence has been added: 'TIR-only retrievals are based on the 5A, 5D, and 7D radiances in the 4.7 micron band, whereas NIR-only retrievals are based solely on the ratio of the 6D and 6A radiances in the 2.3 micron band.'

- Page 2: Figure numbering in the text jumps from 1 to 3; Figure 2 is not mentioned.

Response: This is an error which has been corrected in the revised manuscript - Fig. 2 is now introduced at the beginning of the last paragraph of the Introduction (between the initial discussions of Fig. 1 and Fig. 3).

- Section 2: Please briefly introduce the four aspects of the retrieval enhancement at the beginning of this section.

Response: Such a paragraph has been added at the beginning of Section 2. The paragraph reads: 'As detailed in the sections below, the Version 8 retrieval algorithm incorporates updated spectroscopic information used in the radiative transfer model, improved methods for radiance bias correction and averaging kernel calculations and, finally, the most recent version of the MODIS cloud mask.'

- Page 5 lines 13-17 are not fully clear. Why change the relative channel 7 average radiance in order to obtain consistency between V7 and V8 processing?

Response: What actually changed for V8 was the specified threshold ratio between the 7A calibrated radiance and model-simulated radiance (based on the assumption of clear sky) which is used for cloud testing. A threshold ratio of 1 would be ideal given a perfect clear-sky radiative transfer model. The change in the threshold ratio was required because of changes in the radiative transfer model detailed in Section 2.1. Since we have no evidence that the cloud detection for V7 was either too aggressive or too conservative, we feel that the goal of achieving consistency in cloud detection yield for V7 and V8 is justified.

- Section 3.1: Please briefly elaborate on the "flask sampling system"

Response: Several sentences have been added describing the NOAA in-situ analysis system and measurement accuracy. The following text has been added: 'Typical profiles are derived from a set of twelve flasks. Reproducibility of the laboratory-measured CO dry-air mole fractions, which are measured by either a vacuum UV–resonance fluorescence spectrometer or a reduction gas analyzer is better than 1 ppb.'

- Page 7 lines 1-6 are not fully clear. First it is said that the in-situ measurements are vertically extended, than it is said that validation results for the 100 hPa level are not reported because of a lack of in-situ measurements above this level.

Response: These statements are both correct.  The purpose of vertically extending the in-situ measurements measured from aircraft is to fill in the 'gap' of missing data in x_true (at the top of the profile) required to calculate x_sim = x_a + A(x_true - x_a).  Retrievals of CO at the 100 hPa level (representing the layer from 100 to 50 hPa) cannot be reliably validated, however, because they heavily depend on extended (extrapolated) CO measurements from lower altitudes.  At lower retrieval levels the associated averaging kernels are weighted towards lower altitudes and the sensitivity to the extrapolated CO measurements is much weaker.   The end of the paragraph has been revised and now reads: 'Validation results for the MOPITT 100 hPa retrieval level are not reported below, since apparent retrieval errors due to reliance on the model-based extension at the top of the profile is much greater than for lower retrieval levels.  Reliable validation of the MOPITT 100 hPa retrieval level would require in-situ profiles that reach higher altitudes than are currently available.'

- Page 8 lines 28-29: Does the last sentence of Section 3.4 indicate that all significant bias has been removed and that only 5 % spread remains? This is not fully clear from the current phrasing.

Response: The last sentence just indicates that retrieval biases obtained with the AC and KORUS datasets are generally consistent (to within about 5%) with the biases obtained using the NOAA profiles.

- Figures: The authors should explain in the text how the "bias drift", "slope", and the uncertainties thereon are calculated.

Response: Bias drift calculations are discussed in the second paragraph of Section 3.2: 'Bias drift is calculated as the slope of a least-squares best fit applied to the timeseries data presented in Fig. 5, and converted to units of %/yr.'  The sentence has been revised to provide a reference for the conversion in units to %/yr.  Uncertainties in the slope (drift) are calculated according to the standard method for least-squares calculations, which we believe most readers know well.

---

## Author Comment (AC3) · 24 May 2019

Authors' Responses to Reviewer #3 of 'Radiance-based Retrieval Bias Mitigation for the MOPITT Instrument: The Version 8 Product' by M. Deeter et al.

Original reviewer's comments in blue. Authors' responses in black.

**Replies to Comments of Reviewer #3**

This paper describes version8 of the MOPITT CO retrievals and of its associated data record. Despite being short and technical, the paper deals with an increasingly important aspect of satellite remote sensing, which is the consistency and homogeneity in the time series at level 2. By targeting CO, a so-called indirect climate variable, and relying on the longest time-series available from satellites (almost two decades with MOPITT), this paper is a very useful reference. To achieve this consistency in the CO time series for version 8, an empirical and "dynamical" radiance correction scheme is proposed. The methodology is well described and the results give convincing evidence of its benefit in V8 over previous versions. New validation results using recent aircraft measurement campaigns (from ACRIDICON-CHUVA, KORUS-AQ and Atom) are given as well, and these provide an additional support for the proposed radiance bias minimization scheme. The paper is extremely clear and very well written. For these reasons, I recommend publication of the paper in AMT. I list below a few comments or questions that the authors may want to consider to possibly improve or clarify some aspects of the paper.

The authors appreciate the reviewer's useful comments.

- Page2, discussion around Figure 1: the drifts compared to the measurement flasks are mostly positive below 500 hPA and negative above. In the total columns this results in almost no drift. This could be indicated from the start (around line 25).

Response: The following sentence has been added in the fourth paragraph of the Introduction: 'However, opposing drift in the upper and lower troposphere appears to mostly cancel with respect to the retrieved total column [Deeter et al, 2013].'

- Page 2, line 30: To explain the latitudinal-dependent bias in the V7-TIR only CO, the water vapor field is suspected to be a cause. From Figure 3 it seems indeed obvious that water plays a role and this is one of the reason for the bias correction » suggestion to remove "perhaps" on line 30?

Response: The word 'perhaps' is used since either of two quite different effects (errors in the radiative transfer modeling of water vapor or, alternatively, the accuracy of water vapor profiles in the MERRA-2 reanalysis) could conceivably explain the results.

- Would a figure showing the (global? hemispheric?) averaged retrieved profile from v7 and V8 with the associated standard deviation not be insightful? Is this possible and does this add something?

Response: We think this is an interesting idea, but have not yet pursued it. Ideally, such an analysis should involve V7/V8 comparisons at different stages of the MOPITT mission (to expose the effects of bias drift minimization) and for multiple regions and seasons (to expose the effects of water vapor corrections). Differences between V7 and V8 might also vary with surface type (land or ocean) and thermal contrast (day vs night). Such an analysis would significantly add to the length of the paper.

- Page 5 top; it would be useful to see how to new way to calculate the averaging kernels in v8 affects the averaging kernel calculations. Also, is there an impact on the validation between V7 and V8 that could potentially be related to the AVKs calculations?

Response: The following sentence has been added to the last paragraph of Section 2.3: 'While the new method for calculating the total column averaging kernel is more rigorous than the previously used method, resulting differences in total column validation statistics (correlation coefficient, bias, and standard deviation) were found to be insignificant.

- Page 5, line 10. Remove parentheses for that sentence?

Response: Agreed.

- Page 5, equation 5. Is x_true different than x_cmp from eq. (3)? I understand that x_cmp is more general but for this paper is the distinction useful?

Response: These two terms are really not interchangeable. We begin using x_true starting with Eq. 5 since validation involves comparisons with some type of 'truth' as the reference.  In cases where MOPITT data are used to evaluate models, MOPITT may actually be the reference, and therefore it would be inappropriate to use the term x_true for the model.

- Page 6, last sentence before section 3.1; I would suggest to move this sentence earlier in section 2.3

Response: Agreed.

 - Figure 5 is similar to Figure 1 for the new version. When discussing Figure 5 on page 7 it would be good to refer the reader to Figure 1 for a useful comparison

Response: Agreed.

- Figures 5 and 6 are convincing on what the dynamic bias-correction brings. Nice results; well Done.

Response: Thanks!

- Figure 8 compares the new CO profiles from V8 to more recent reference measurements from aircrafts. As said before, this is useful and again convincing. It would, however, be nice to show (or at least comment) how V7T performs compared to these reference dataset (in comparison to V8T).

Response: We feel that the V7/V8 comparisons using the NOAA and HIPPO datasets are sufficient to establish the improved performance of the V8 product, which is the main subject of the manuscript.

- Page 8. The last sentence before section 3.3 refers to biases at high northern latitude but looking at Figure 8 one has the feeling that there are almost no measurements in the latitude band 60-90 N. Is that the case and could you please then indicate how significant this bias is?

Response: We agree that the sparseness of in-situ profiles at high northern latitudes during ATom decreases the statistical significance of the corresponding validation results - the large error bars for the latitude band 60N-90N in Fig. 8 may well be the result of a small number of 'outliers'.  We have therefore revised the text in Section 3.2 to read: 'Biases outside of this range are most evident between

60 N and 90 N.  However, this could be related to the sparseness of profiles in this region and the influence of a small number of outliers.'

- Comparing Figure 12 and Fig8 one sees that the biases in v8 are larger for the TIRNIR retrievals than for the TIR only. Was this feature also there for v7? Is there a known reason for this?

Response: Both systematic errors and retrieval noise tend to be larger in the TIR-NIR product due to the strategy to amplify the weight assigned to the NIR measurements.  Text explaining this point, along with an additional reference, has been added to Section 3.4.

- Check if the reference Worden et al., 2013 is cited in the text.

Response: This reference should have been cited at the end of the fourth paragraph of the Introduction. This has been corrected.

- Caption of table 2: correlation coefficients are given between retrieved quantities and corresponding a priori quantities. Is it a priori or reference?

Response: That statement in the caption is meant to clarify that correlations are calculated for the quantities ($x\_rtv$ - $x\_a$) vs. ($x\_sim$ - $x\_a$), rather than $x\_rtv$ vs $x\_sim$.  Subtracting the a priori from $x\_rtv$ and $x\_sim$ before making the correlation calculation eliminates 'false correlations' that are just related to variability in the a priori.  A sentence has been added at the end of the first paragraph in Section 3.2 to clarify this point: 'Correlations due simply to the variability of the a priori are avoided by basing correlation coefficient calculations on ($x\_rtv$ - $x\_a$) rather than $x\_rtv$.'  The issue is no longer discussed in the caption.

- Would it not be possible to make a single table out if Tables 4-6? This would be nice and also helpful to easily compare the results from each validation set.

Response: Agreed.

---

## Author Response (AR1)

**Radiance-based Retrieval Bias Mitigation for the MOPITT Instrument: The Version 8 Product**

Merritt N. Deeter1, David P. Edwards1, Gene L. Francis1, John C. Gille1, Debbie Mao1, Sara Martínez-Alonso1, Helen M. Worden1, Dan Ziskin1, and Meinrat O. Andreae2,3

[revised manuscript text omitted]

---

## Author Response (AR2)

Authors' Responses to Associate Editor's Comments on June 18, 2019 of 'Radiance-based Retrieval Bias Mitigation for the MOPITT Instrument: The Version 8 Product' by M. Deeter et al.

Original Editor's comments in blue.  Authors' responses in black.

**Replies to Associate Editor's Comments**

Reviewer # 1:
Item (1): Although I completely accept that comparison to TCCON data is a work on its own and will be published in a forthcoming paper, I think it would be appropriate to mention the existence of these data, the reasoning why it has not been included here, and the future plan to validate the MOPITT data against those data as well. This could go either into the introduction (which I would prefer) or into Chapter 3 (Validation).

Response: The following text has been added in the first paragraph of Section 3. "Other remote sensing datasets, such as the TCCON ('Total Carbon Column Observation Network') and NDACC ('Network for the Detection of Atmospheric Composition Change') datasets  are potentially useful for MOPITT validation [Buchholz et al., 2017], however results are more difficult to analyze due to differences in retrieval averaging kernels and a priori [Rodgers and Connor, 2003]. Thus, results presented in this paper are solely based on aircraft in-situ profiles whereby averaging kernel effects are taken fully into account."  Note the two added references.

Item (2): I have the strong impression that including (some of) the discussion on the topic of stability of measurements, e.g. your responses to the reviewer's concerns, would considerably strengthen the paper. I understand that you do not want to speculate on the sources of the drifts as long as they have not clearly been indentified, and I accept this decision. However, the discussion of the concerns wrt an ad-hoc long-term-corrections and your justification why to do it would help to convince a potential reader having the same concerns as the reviewer. I would recommend to include parts of your arguments in an appropriate way in the conclusions.

Response: We have added the following paragraph to the Conclusion.  "Our chosen strategy to rely on the stability of the NOAA aircraft in-situ data to determine the time-dependent radiance bias correction factors does imply that the fidelity of CO trends in the radiance bias-corrected MOPITT data (i.e., the V8 retrieval product) is ultimately limited by the stability of the NOAA in-situ measurements. However, the NOAA measurements are widely accepted as a standard for long-term CO analyses, and are calibrated using the World Meteorological Organization (WMO) mole fraction scale [Sweeney et al., 2015]. The NOAA dataset is therefore well qualified for climate analyses.  On the other hand, we acknowledge that bias drift in the V8 product for regions not represented by the NOAA aircraft network (primarily covering North America) could be substantially different.  This seems unlikely though, particularly if the source of the bias is instrumental."

Finally, I have a number of comments on my own:

Abstract: The abstract is too short and does not provide necessary details. At least it should be mentioned against which data sets the validation has been performed, and what the outcome was in numbers. The statement "Validation results illustrate clear improvements with respect to long-term bias drift and geographically variable retrieval bias." is not sufficient and not appropriate for a peer-reviewed journal paper.

Response:  The revised abstract now lists all of the datasets used for validation.  However, because of the need to provide the complete (non-abbreviated) title for each dataset/campaign, this revision adds greatly to the length of the abstract.  The cited sentence ("Validation results illustrate ...") provides a high-level summary of the validation results; it is not obvious to us what makes this sentence inappropriate.  To elaborate on this sentence, we have added the following text. "For example, whereas bias drift for the V7 TIR-only product exceeded 0.5 %/yr for levels in the upper troposphere (e.g., at 300 hPa), bias drift for the V8 TIR-only product is found to be less than 0.1 %/yr at all levels. Also, whereas upper-tropospheric (300 hPa) retrieval bias in the V7 TIR-only product exceeded 10 % in the tropics, corresponding V8 biases are generally less than 5 % (in terms of absolute value) at all latitudes and do not exhibit a clear latitudinal dependence."

Introduction:
The introduction misses to a certain part putting the presented work into a larger context. It is done wrt the previous data versions, i.e. for version 7, of MOPITT. However, I miss to some degree a discussion of available data sets suited for validation, and some reasoning why the validation presented here focusses on the Aircraft in-situ data. At the end of the introduction, a sentence or two should be added about the structure of the paper, i.e. what is dealt with in which chapter.

Response: The text added to Section 3 in response to the Editor's first comment above addresses the reason for relying on in-situ profiles for validation.  The following text has been added at the end of the Introduction. "The remaining sections of this manuscript describe revisions made to the MOPITT Version 8 retrieval algorithm (Section 2), the validation methodology and results (Section 3) and, finally, the conclusions drawn from the results (Section 4)."

Fig. 3 and Fig. 7: Please explain the colour code in the figure caption.

Response:  The revised captions for Figures 3 and 7 now both include the sentence 'Colors indicate the particular phase of the HIPPO mission, as described in the caption to Figure 2.'

List of Changes made to the Manuscript

1. Abstract expanded to include titles of validation datasets.
2. Abstract expanded to provide examples of product improvements.
3. Introduction now includes ending sentence which outlines the remaining sections.
4. Section 3 now mentions TCCON and NDACC datasets and explains reason for relying on CO in-situ vertical profiles measured from aircraft.
5. Conclusion includes new paragraph explaining strategy for reducing bias drift in V8 products.
6. Captions to Figs. 3 and 7 now refer to color-coding scheme described in Fig. 2 caption.

[revised manuscript text omitted]